# A critical role for heme synthesis and succinate in the regulation of pluripotent states transitions

**Damien Detraux[1,2], Marino Caruso[1], Louise Feller[1], Maude Fransolet[1], Sébastien Meurant[1], Julie Mathieu[2,3], Thierry Arnould[1], Patricia Renard[1]\***

[1]Laboratory of Biochemistry and Cell Biology (URBC), NAmur Research Institute for LIfe Sciences (NARILIS), University of Namur (UNamur), Namur, Belgium, Namur, Belgium; [2]Institute for Stem Cell and Regenerative Medicine, University of Washington, Seattle, United States; [3]Department of Comparative Medicine, University of Washington, Seattle, United States

**Abstract** Using embryonic stem cells (ESCs) in regenerative medicine or in disease modeling requires a complete understanding of these cells. Two main distinct developmental states of ESCs have been stabilized in vitro, a naïve pre-implantation stage and a primed post-implantation stage. Based on two recently published CRISPR-Cas9 knockout functional screens, we show here that the exit of the naïve state is impaired upon heme biosynthesis pathway blockade, linked in mESCs to the incapacity to activate MAPK- and TGFβ-dependent signaling pathways after succinate accumulation. In addition, heme synthesis inhibition promotes the acquisition of 2 cell-like cells in a heme-independent manner caused by a mitochondrial succinate accumulation and leakage out of the cell. We further demonstrate that extracellular succinate acts as a paracrine/autocrine signal, able to trigger the 2C-like reprogramming through the activation of its plasma membrane receptor, SUCNR1. Overall, this study unveils a new mechanism underlying the maintenance of pluripotency under the control of heme synthesis.

\*For correspondence:
patsy.renard@unamur.be

**Competing interest:** The authors declare that no competing interests exist.

## Editor's evaluation

In their study, Detraux D and colleagues provide compelling evidence demonstrating a role for heme biosynthesis on FGF-ERK and TGF β signalling and exit from naïve pluripotency, and in controlling the 2-cell-like cell state. The observations and conclusions provided by the authors are convincing and potentially relevant in the field of pluripotent cell state transitions.

## Introduction

The development and regeneration of an organism are two processes held by stem cells. These cells possess unique features such as an unlimited capacity for self-renewal and the ability to differentiate into various cell types. With their pluripotent phenotype, embryonic stem cells (ESCs) retain the ability to differentiate into all cell types of the embryo. First in mouse (*Martin, 1981*; *Evans and Kaufman, 1981*), then later in human (*Thomson et al., 1998*), two main states of pluripotent stem cells have been described: the naïve ESCs, resembling the inner cell mass (ICM) of the pre-implantation embryo epiblast, and the primed ESCs, mirroring the epiblast of the post-implantation stage. While these cells represent timely close stages in embryo development, they display dramatic differences, such as developmental potential (*Gafni et al., 2013*; *Ware et al., 2014*), epigenetic landscape, X-chromosome inactivation pattern and metabolic activity (*Sperber et al., 2015*; *Theunissen et al., 2016*; *Zhou*

*et al., 2012*; *Sahakyan et al., 2017*; *Grow et al., 2015*; *Takashima et al., 2014*; *Nichols and Smith, 2009*). Aside from these two pluripotent stages, ESC culture is known to be very heterogenous in terms of pluripotent or epigenetic marker expression (*Sustáčková et al., 2012*). Interestingly, a small population (representing about 1 %) of mouse ESCs grown in naïve conditions displays features of the two-cell stage embryo (2C-like population or 2CLCs) exhibiting extended potential (*Macfarlan et al., 2012*). This subset also displays the expression of 2C-specific genes such as the *Zscan4* cluster, the retro-transposable element *MuERVL* or the master regulator DUX (*Macfarlan et al., 2012*; *Percharde et al., 2018*; *Ishiuchi et al., 2015*), a disappearance of chromocenters and loss of the core pluripotency protein OCT4 (*Macfarlan et al., 2012*). However, despite the well-known differences between the different pluripotent states, little is understood about the molecular mechanisms governing the transition between them.

Several studies have started to address the question of the naïve-to-primed ESC transition using different screening methods to identify genes controlling this transition (reviewed in *Li et al., 2020*). Notably, a few studies have revealed mTORC1/2 as a critical component for the naïve-to-primed transition (*Mathieu et al., 2019*; *Li et al., 2018*; *Duggal et al., 2015*) that regulates key developmental pathways such as Wnt signaling (*Sperber et al., 2015*; *Mathieu et al., 2019*; *Xu et al., 2016*; *Taelman et al., 2019*). Among the hits of the different screens, we observed the recurrence of genes involved in the heme biosynthesis pathway. This pathway, starting in mitochondria, uses succinyl-CoA and glycine as starting material. It then proceeds to successive cytosolic reactions before ending by the formation of the heme molecule in the mitochondrial matrix. Although largely studied in hematopoietic stem cells, the roles of this biosynthetic pathway and this metabolite have never been studied in the context of pluripotency. In this study, we demonstrate the incapacity of mouse ESCs (mESCs) to exit from naïve toward the primed state under heme synthesis inhibition, a process caused by the inability to activate the core MAPK and TGFβ-SMADs signaling pathways due to an accumulation of extramitochondrial succinate. We also show that heme synthesis inhibition in naïve mESCs favors the emergence of 2CLCs in the cell population. This effect is heme-independent as hemin supplementation does not prevent it. We next demonstrate that the reprogramming in 2C-like state upon heme biosynthesis inhibition is actually caused by the accumulation and release of succinate derived from the heme synthesis inhibition, acting as both paracrine and autocrine signals.

## Results

### Murine ESCs are dependent on heme biosynthesis to properly transition to the primed stage

In order to unveil the pathways required for the naïve-to-primed ESC transition, we compared the results of whole genome CRISPR-Cas9 screens previously published for mouse and human ESCs (*Mathieu et al., 2019*; *Li et al., 2018*). The significant hits (p val <0.05) were submitted to the DAVID functional annotation tools. For the human screen, positive hits for apoptosis were removed. Indeed, since a negative selection was applied to induce the death of primed human ESCs (hESCs), cells that acquired a resistance to cell death by mutating genes involved in apoptosis would be spared. *Figure 1a* displays the top Gene Ontologies (GO) for the DAVID biological processes showing 'heme biosynthetic process' as one of the most enriched in both studies (*Mathieu et al., 2019*; *Li et al., 2018*). Seven out of 8 enzymes of this metabolic pathway (ALAD, PBGD, UROS, UROD, CPOX, PPOX, and FECH) came out as positive hits in the CRISPR screen during the naïve-to-primed mESC transition. In hESCs, only the 4 cytosolic enzymes (PBGD, UROS, UROD, and CPOX) were highlighted. In both models, the expression of these genes both in vitro and in vivo is not modified *Table 1*. Together, the results stress the importance of this metabolic pathway for the naive-to-primed transition, although its role in non-hematopoietic stem cells is poorly understood.

Experimentally, to trigger the exit of the naïve mESC state, the serum-free media with naïve cytokines cocktail (LIF; CHIR99021 and PD0325901; 2iL) is switched to a media with fibroblast growth factor 2 (FGF2) and activin A for 48 hr, allowing mESCs to gain post-implantation features (EpiLC). Using succinylacetone (SA) to inhibit heme synthesis, by interfering with ALAD activity (*Sassa and Kappas, 1983*), and 10 µM of hemin for its rescue, we show that, when mESCs are pushed for 48 hr to exit the naïve stage (EpiLC) in the presence of SA (EpiLC +SA), the expression of the primed gene markers *Fgf5*, *Fgf15*, *Otx2*, *Oct6*, *Dnmt3a*, and *Zic2* is significantly reduced (*Figure 1b*). In addition,

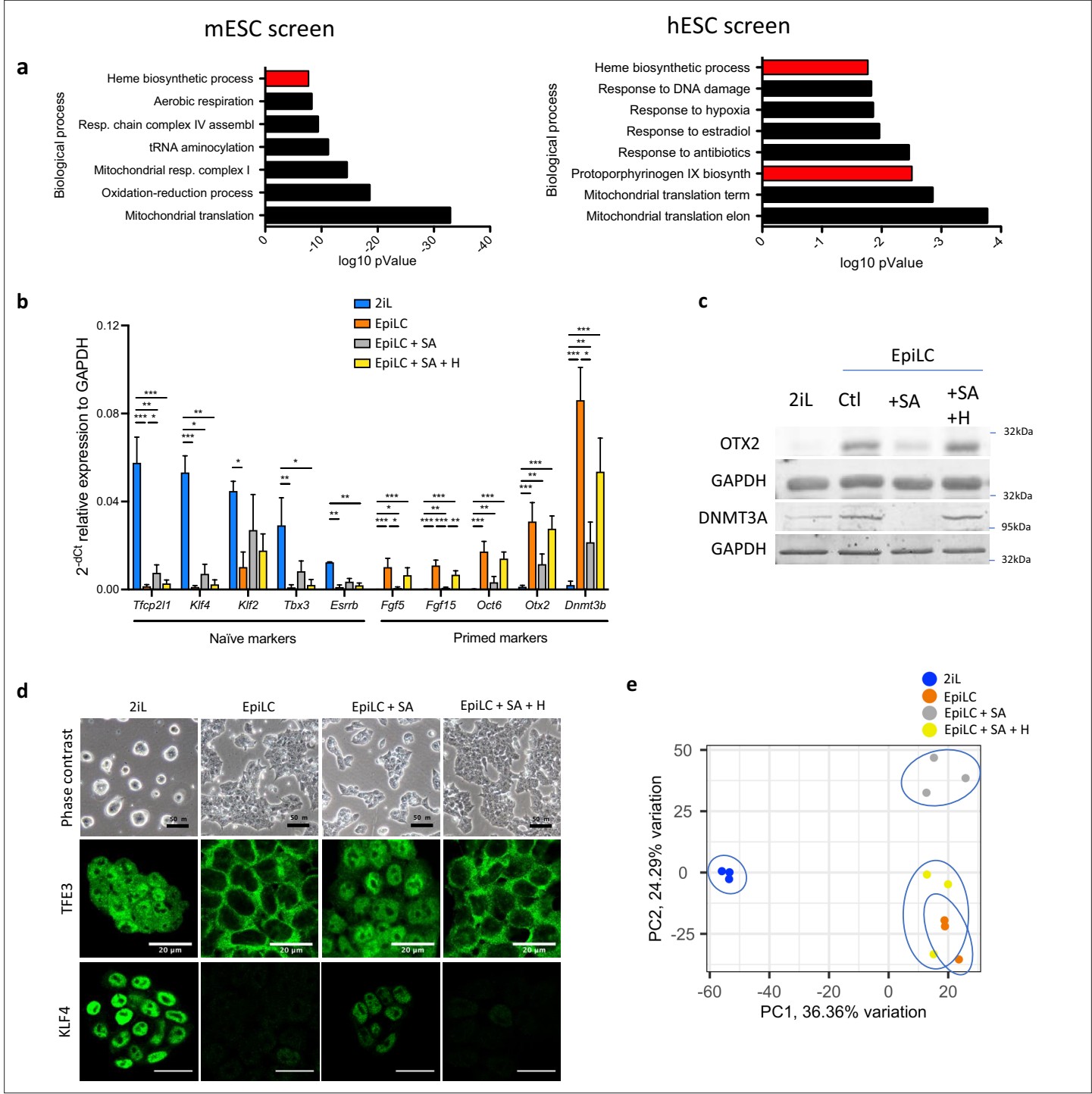

**Figure 1.** Heme synthesis inhibition impairs the exit of mESCs from the naïve state; effect mediated by heme. (**a**) DAVID biological processes GO enrichment from two independent CRISPR-Cas9 screens for the naive state exit, in mouse (left panel) (***Li et al., 2020***) and in human (right panel) (***Mathieu et al., 2019***). The heme biosynthetic pathway is highlighted In red. (**b**) Relative expression of naïve and primed markers of mESCs in naïve conditions (2iL), in transition for 2 days to the Epi stage (EpiLC) with or without 0.5 mM succinylacetone as heme synthesis inhibitor (EpiLC +SA) and 10 µM hemin supplementation (EpiLC +SA + H), assessed by RT-qPCR relative to *Gapdh* expression (*Tfcp2l1*, transcription factor CP2-like 1; *Esrrβ*, estrogen-related receptor β; *Klf2/4*, Kruppel-like factor 2/4; *Tbx3*, T-Box Transcription Factor 3; *Fgf5/15*, fibroblast growth factor-5/15; *Zic2*, zic family member 2; *Otx2*, homeobox protein 2). Results expressed as mean +/-S.D. *p<0.05, **p<0.01, ***p<0.001. ANOVA-1. n=3 independent biological replicates. (**c**) Western blot analysis of the protein abundance of OTX2 and DNMT3A relative to GAPDH as a loading control for cells in naïve conditions (2iL), in transition for 2 days to the Epi stage (EpiLC) with 0.5 mM SA (+SA) and 10 µM hemin (+H) supplementation. Representative blot of three

*Figure 1 continued on next page*

*Figure 1 continued*

biological replicates. (**d**) Phase contrast micrographs of cells in naive (2iL), or in transition to the primed (EpiLC) state with treatment with SA and hemin (**H**). Scale bar = 50 µm. Confocal micrographs of mESCs in naive stage or in transition for TFE3 (Transcription Factor Binding To IGHM Enhancer 3) and KLF4, in green. Scale bar = 20 µm. TFE3 n=3 and KLF4 n=3 biological replicates (**e**) Principal component analysis (PCA) of the normalized RNAseq data transcripts.

The online version of this article includes the following source data and figure supplement(s) for figure 1:

**Figure supplement 1.** Heme synthesis inhibition perturbs the exit of the naïve state in mESCs and hESCs.

**Figure supplement 1—source data 1.** Raw uncropped and annotated western blot images for *Figure 1—figure supplement 1c*.

**Source data 1.** Raw uncropped and annotated western blot images for *Figure 1c*.

the loss of expression for naïve markers (*Esrrb, Tfcp2l1, Klf2-4, and Tbx3*) is partially prevented in these conditions. This was confirmed at the protein level by a decrease in the abundance of OTX2 and DNMT3A analyzed by western blot (*Figure 1c*) and the increase in the abundance of KLF4 by immunofluorescence, when cells were treated with SA during the transition (*Figure 1d*). Furthermore, the subcellular localization of TFE3, mainly nuclear in naïve cells and only cytosolic in primed cells (*Mathieu et al., 2019*; *Betschinger et al., 2013*), remains nuclear in the presence of SA (*Figure 1D*). Hemin supplementation restores the gene expression, the protein abundance and the subcellular localization of TFE3 to levels similar to those found in cells incubated without SA (*Figure 1a–d*). To consolidate this, we then generated an ALAD KO line, which was maintained in culture with hemin supplementation to avoid degeneration. In accordance with the results obtained with SA, removal of hemin during the transition perturbed the exit from the naive state (*Figure 1—figure supplement 1*). Finally, principal component analysis (PCA) of the normalized gene expression from RNA sequencing also reveals the segregation of the cells treated with SA (EpiLC + SA) from either the controls (EpiLC) or the cells rescued with hemin (EpiLC +SA + H) (*Figure 1e*). An impairment of the naïve state exit is also observed to some extent in human ESCs (Elf1 cell line) when pushed to exit in TeSR for 4 days (*Figure 1—figure supplement 1c*). Overall, these results confirm the functional screen data by showing that the inhibition of heme biosynthesis impairs the naive-to-primed mESC transition.

## Heme synthesis inhibition prevents the activation of key signaling pathways associated with implantation

Heme deficiency is canonically sensed in mammals through the nuclear factor BACH1 (reviewed in *Zhang et al., 2018*), increasing its nuclear localization or the activation of the integrated stress response (ISR; reviewed in *Pakos-Zebrucka et al., 2016*) through the action of the heme-regulated inhibitor (HRI). While we did not observe an accumulation of BACH1 in the nucleus following SA treatment (*Figure 2—figure supplement 1a*), we did observe an increase in EIF2α phosphorylation and a reduction in global protein translation, both showing activation of the ISR (*Figure 2—figure supplement 1b-c*). However, activation of this pathway with a chemical activator of HRI, BTdCPU (*Chen et al., 2011*), effectively reducing protein translation to the extent of SA, did not prevent the exit from the naive stage (*Figure 2—figure supplement 1d-e*). Thus, to identify the mechanisms involved in the failure of the cells to properly undergo the transition, we performed a gene set enrichment analysis (GSEA) with the KEGG pathways (Kyoto Encyclopedia of Genes and Genomes) between EpiLC and EpiLC +SA RNAseq data. While genes involved in xenobiotic detoxification or cytochrome P450 are expected to be upregulated in response to SA-induced heme deprivation (*Vinchi et al., 2014*; *Lämsä et al., 2012*), many crucial signaling pathways involved in development are shown negatively enriched in SA-treated condition (*Figure 2a*). We thus focused our attention on the pathways directly associated with the naive-to-primed transition that is triggered by the combined presence of FGF2 and activin A in the growth culture medium, especially since detailed analysis of the PC loadings driving the PC2 separation (*Figure 1e*) highlights several SMAD pathway-related proteins (*Figure 2—figure supplement 1f*). On the one hand, the MAPK-ERK1/2 pathway (downstream of FGF2) was not activated in EPI +SA cells, as shown by the absence of phosphorylation of ERK1/2 (*Figure 2—figure supplement 1g*). On the other hand, the activin A-SMAD pathway activation was also compromised as shown by the difference in nuclear localization of SMAD2/3 when compared to the EpiLC cells (*Figure 2—figure supplement 1h*) or SMAD3 phosphorylation (*Figure 2b–c*). Interestingly, chemical inhibition of those two pathways by 1 µM PD0325901 (MEKi) and 5 µM SIS3 (TGFβ-SMADi)

**Table 1.** Relative expression of heme synthesis enzymes.

(a) Fold change expression of the 8 enzymes of the heme synthesis pathway, based on normalized data from bulk RNAseq or proteomics analysis, extracted from published datasets. (b) Normalized gene expression counts of heme synthesis enzymes from in vivo mouse blastocysts analyzed by RNAseq.

**a**

| Source | Di Stephano et al. | Di Stephano et al. | Grow et al. | Sperber et al. | Nakamura et al. | This study |
|---|---|---|---|---|---|---|
| | **Human ESCs** | | | | **Mouse ESCs** | |
| Type | Proteomics | Transcriptomics | Transcriptomics | | | |
| ALAS | 0,7 | 0,88 | 1,22 | 0,95 | 0,85 | 1,016 |
| ALAD | 0,47 | 0,29 | 0,81 | 0,96 | 0,54 | 0,92 |
| HMBS | 1,43 | 0,84 | 1,22 | 1,27 | 1,15 | 0,81 |
| UROD | 1,12 | 1,05 | 0,62 | 1,10 | 1,33 | 0,64 |
| UROS | 2,09 | 1,58 | 0,78 | 1,12 | 0,50 | 0,75 |
| CPOX | 1,24 | 2,07 | 1,02 | 1,25 | 1,12 | 0,87 |
| PPOX | 2,09 | 1,54 | 1,16 | 1,37 | 1,25 | 0,56 |
| FECH | 0,98 | 1,10 | 1,28 | 0,77 | 1,06 | 1,30 |

**b**

| | | Mouse blastocysts | | |
|---|---|---|---|---|
| | | Nakamura et al. | | |
| Enzyme | Stage | E4.5 | E5.5 | E6.5 |
| | Alas | 17 | 12 | 16 |
| | Alad | 40 | 22 | 33 |
| | Hmbs | 73 | 65 | 60 |
| | Uros | 2,2 | 5,4 | 5,7 |
| | Urod | 43 | 53 | 80 |
| | Cpox | 22 | 13 | 14 |
| | Ppox | 14 | 7 | 7 |
| | Fech | 8 | 26 | 28 |

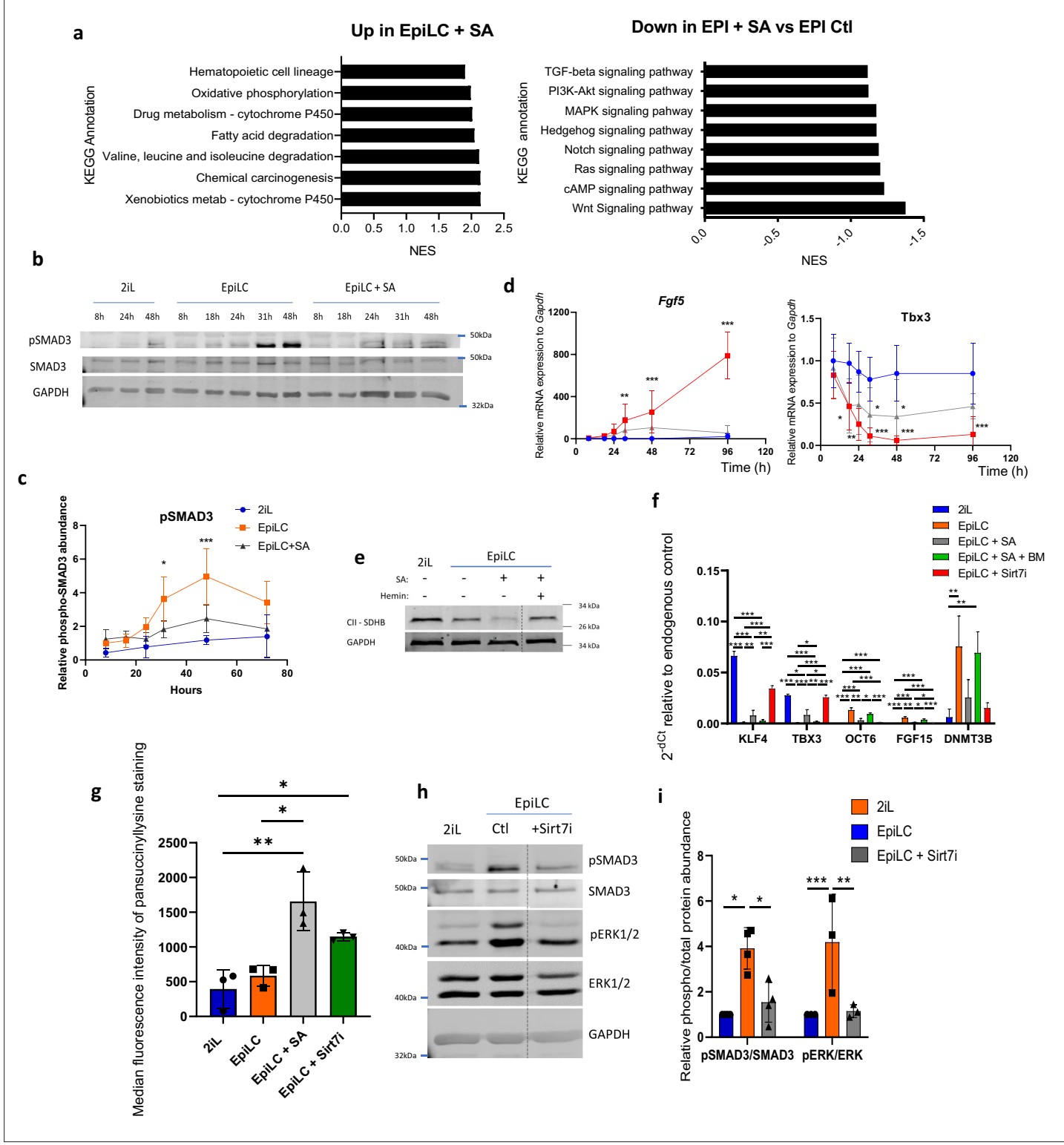

**Figure 2.** SA prevents the activation of the MAPK and Activin A-SMAD pathways during the mESC transition through the cytosolic accumulation of succinate. (**a**) GSEA performed on RNAseq data were analyzed for KEGG pathways. Up- and Down-regulated KEGG pathways in EpiLC + SA versus EpiLC Ctl, represented as normalized enrichment scores (NES). (**b**) Western blot analysis of the time course protein abundance of SMAD3 and phospho-SMAD3 relative to GAPDH as a loading control for cells in naïve conditions (2iL), in transition for up to 2 days to the Epi stage (EpiLC) with 0.5 mM SA as heme synthesis inhibitor. Representative blot of 5 biological replicates, and quantified in (**c**). (**d**) Time course gene expression analysis of Fgf5 and Tbx3 assessed by RT-qPCR, expressed as mean +/-S.D. ANOVA-2. n=5 independent biological replicates. (**e**) Western blot analysis of SDHB during the

*Figure 2 continued on next page*

*Figure 2 continued*

transition, relative to GAPDH as loading control. Representative blot of 2 biological replicates. (f) Relative expression of naïve and primed markers of mESCs in naïve conditions (2iL), in transition for 2 days to the Epi stage (EpiLC) with or without 0.5 mM SA (EpiLC + SA), 1 µM butylmalonate (+BM) or 5 µM of Sirt7 inhibitor supplementation, assessed by RT-qPCR relative to *Gapdh* expression. n=3 independent biological replicates. ANOVA1 (g) Median fluorescence intensity of succinyllysine residues measured by flow cytometry for mESC in transition with 5 µM Sirt7 inhibitor (Sirt7i). n=3 independent biological replicates, ANOVA1. (h) Representative western blot analysis of the abundance of the total and phosphorylated forms of SMAD3 and ERK1/2, using GAPDH as loading control, for mESC in transition with a Sirt7 inhibitor (Sirt7i), and quantified in (i).

The online version of this article includes the following source data and figure supplement(s) for figure 2:

**Source data 1.** Raw uncropped and annotated western blot images for *Figure 2b*.

**Source data 2.** Raw uncropped and annotated western blot images for *Figure 2e*.

**Source data 3.** Raw uncropped and annotated western blot images for *Figure 2h*.

**Figure supplement 1.** SA prevents the activation of the MAPK and Activin A-SMAD pathways during the mESC transition.

**Figure supplement 1—source data 1.** Raw uncropped and annotated western blot images for *Figure 2—figure supplement 1b*.

**Figure supplement 1—source data 2.** Raw uncropped and annotated western blot images for *Figure 2—figure supplement 1c*.

**Figure supplement 1—source data 3.** Raw uncropped and annotated western blot images for *Figure 2—figure supplement 1d*.

**Figure supplement 1—source data 4.** Raw uncropped and annotated western blot images for *Figure 2—figure supplement 1g*.

(*Figure 2—figure supplement 1g-h*) shows that the inhibition of the SMAD translocation mimics the transition defects observed with SA as revealed by the gene expression analysis of naive and primed markers (*Figure 2—figure supplement 1i*). This pathway inhibition was especially observed after 24 h of treatment, and concomitant with the defect in gene expression (*Figure 2b–d*). Addition of two- or threefold increased doses of activin or FGF2 did not rescue the gene expression pattern, indicating a strong inhibition (*Figure 2—figure supplement 1j*). Mechanistically, previous reports have shown that heme biosynthesis consumes a lot of the glycine and succinyl-CoA precursors in the mitochondria, acting as some sort of 'succinyl-CoA sink' (*Atamna, 2004*). We thus hypothesized that heme synthesis inhibition would increase the abundance of succinyl-CoA in mitochondria, that could then exit the organelle in the form of succinate and accumulate in other cell compartments (*Atamna, 2004*). The accumulation of succinate precursor is reinforced by the observed decreased abundance of the succinate dehydrogenase (SDH), consuming succinate in the TCA, known to be destabilized by the loss of heme (nicely reviewed in *Kim et al., 2012*; *Figure 2e*). Together, these results indicate a possible accumulation of succinate in response to heme synthesis inhibition, that could potentially cause the transition defect. To test this hypothesis, we used butylmalonate (BM), an inhibitor of the dicarboxylate carrier (DIC; SLC25A10), to prevent leakage of succinate from the mitochondrial matrix to the cytosol (*Mills et al., 2018*), and this nicely resumed the transition as shown by the gene expression profile of mESCs in the EpiLC + SA + BM condition (*Figure 2f*).

Succinate accumulation and/or SDH ablation is known to increase protein lysine succinylation (*Smestad et al., 2018*), an effect that is observed upon SA treatment and that we mimicked by inhibiting Sirt7, a cytosolic and nuclear desuccinylase (*Figure 2g*). The ability of this molecule to mimic SA in the pathway's inhibition (*Figure 2h–i*) and to prevent the exit of the naive state (*Figure 2f*) points at succinylation events as the cause of the transition defects.

## Blockade of heme synthesis in naive mESCs triggers the activation of a 2C-like program

In addition to this defect in the exit from the naive state, we found that treatment of naïve 2iL cells with SA also modifies the global gene expression as 2iL + SA samples cluster away from 2iL control cells when analyzed by PCA performed on RNAseq data (*Figure 3a*). Our attention was thus drawn on markers reported to be expressed in the two-cell embryo. Indeed, a small proportion of the mESC population actually expresses a gene signature reminiscent of the 2 C stage, these cells thus called 2C-like cells (2CLCs) (*Macfarlan et al., 2012*). Indeed, the expression of the 142 genes previously identified as upregulated in this 2C-like population (*Macfarlan et al., 2012*) was statistically upregulated in 2iL versus 2iL + SA (*Figure 3b*) and this was further confirmed by RT-qPCR for a selection of the most common markers (*Figure 3c*) and further supported by the upregulation of the 2CLC markers in ALAD KO mESCs (*Figure 3d*). As the studies that characterize this 2C-like population

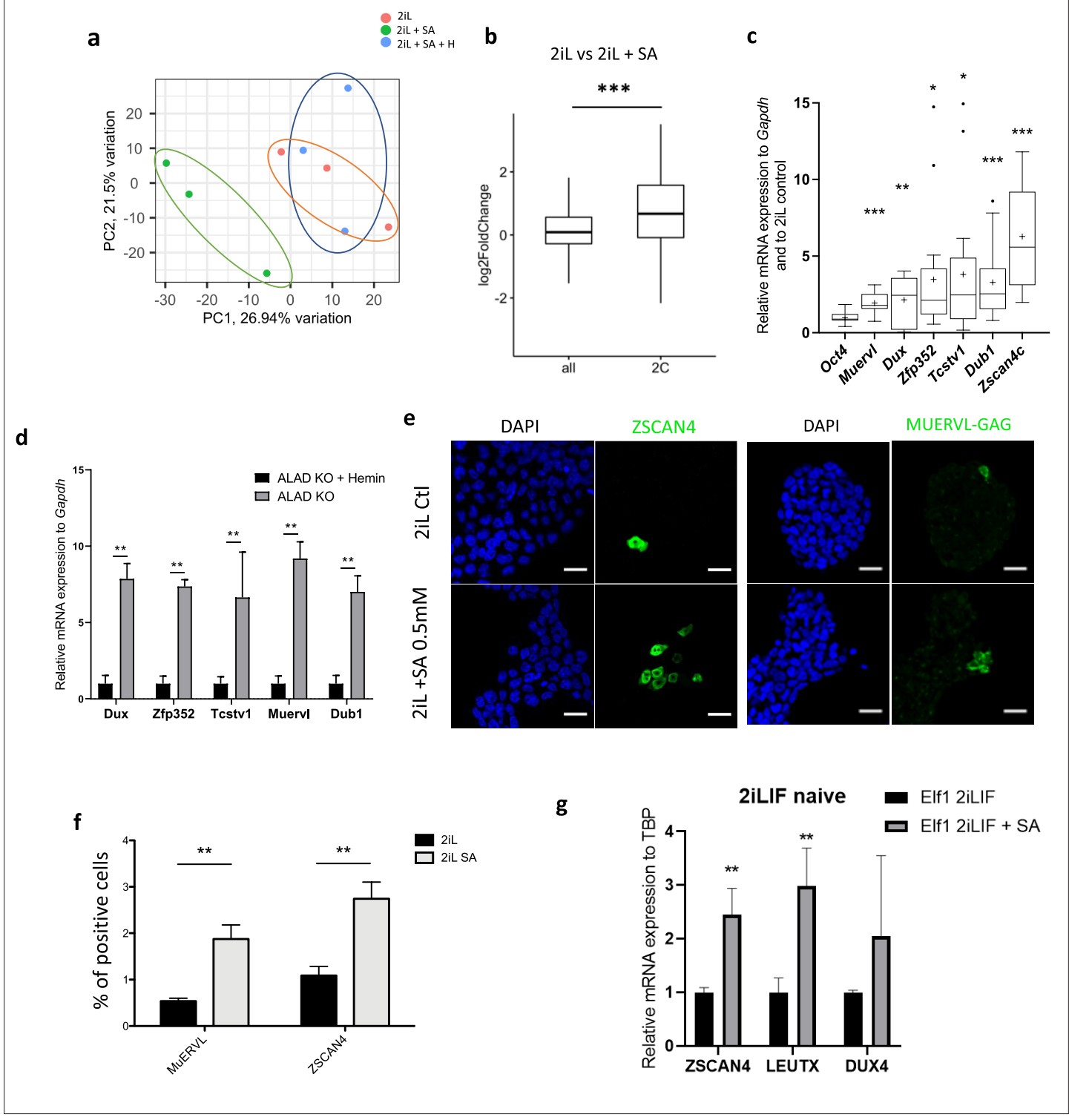

**Figure 3.** Heme synthesis inhibition pushes mESCs toward a 2C-like stage. (**a**) Principal component analysis (PCA) of the normalized RNAseq data transcripts of naïve mESCs (2iL) treated for 48 hr with 0.5 mM SA ± 10 μM Hemin. (**b**) Boxplot of mean Log2FC of 2 C markers defined in *Macfarlan et al., 2012* or all analyzed mRNAs in 2iL +SA versus 2iL control cells. Statistical significance is calculated by Student T-test. p<0.001. (**c**) Relative expression of 2 C gene markers of mESCs assessed by RT-qPCR relative to *Gapdh* expression and to 2iL naive control represented as a Tukey box and whisker plot. The line represents the median and the +is the mean. *Oct4*=octamer-binding transcription factor 4, Muervl = murine endogenous retrovirus-like, Dux = double homeobox, *Zfp352*=Zinc-finger protein 352, *Tcstv1*=2-cell-stage variable group member 1, *Dub1*=Ubiquitin Specific Peptidase 36, *Zscan4c*=Zinc Finger And SCAN Domain Containing 4, isoform c. n=9. (**d**) Gene expression analysis of 2CLC markers of ALAD KO mESCs

*Figure 3 continued on next page*

*Figure 3 continued*

grown for 2 days with or without 10 μM Hemin, assessed by RT-qPCR. n=3 biological replicates. ANOVA1 (**e**) Immunostaining of ZSCAN4 or MUERVL-GAG in untreated naive cells (2iL control) or treated with SA at 0.5 mM. DAPI is used as a nuclear counterstain. Scale bar = 20 μm. (**f**) Percentage of MUERVL- or ZSCAN4-positive cells in the whole population of naïve (2iL) mESCs or naïve treated or not with SA (2iL SA), counted from confocal micrographs as in (**f**) with 10 images per condition for at least 1000 cells per condition. n=4 independent biological replicates. Results expressed as mean +/-S.D. ** p<0.01; T-Tests. (**g**) Relative expression of ZGA related genes in hESCs assessed by RT-qPCR relative to TBP expression and normalized to naive 2iL +IGF + FGF2 (2iLIF) control, with or without 0.5 mM of SA. n=3 independent biological replicates. Results expressed as mean +/-S.D. * p<0.05, **p<0.01, ***p<0.001; t-tests.

report an important heterogeneity at the population level (*Macfarlan et al., 2012*; *Eckersley-Maslin et al., 2016*), we quantified the fraction of ZSCAN4$^+$ and MUERVL-GAG$^+$ cell populations by immunostaining followed by confocal microscopy observations. As shown in *Figure 3e and f*, treatment with SA increases the fraction of 2C-like cells by two to three folds. A similar trend is observed for genes involved in the ZGA for hESCs (*Taubenschmid-Stowers et al., 2022*; *Figure 3g*).

## The induction of the 2C-like program is succinate-dependent

Interestingly, and as opposed to the naive-to-primed setup, the observed phenotype seems independent of heme as it is not rescued by hemin supplementation (*Figure 4a–b*). We thus investigated the putative role of succinate accumulation in the phenotype. The abundance of succinylated proteins in naive cells treated or not with SA was assessed using a pan-succinyllysine antibody in confocal microscopy and in flow cytometry (*Figure 4c*). In basal conditions, the bulk of succinyllysine modifications are located in the mitochondria (*Figure 4—figure supplement 1*), as expected. However, a dramatic increase in the signal associated with succinylated proteins is observed in all subcellular compartments when heme synthesis is inhibited, an effect that is not (or very partially) rescued upon hemin supplementation (*Figure 4c*). We then postulated that blocking the exit of this metabolite from mitochondria would prevent the acquisition of widespread succinyl-lysine post-translational modifications and impair the acquisition of the 2C-like cells markers, only if this phenotype is dependent on increased succinate concentration. As hypothesized, addition of BM combined to SA is correlated to a rescue of both the increase in 2 C markers and the proportion of ZSCAN4 or MUERVL-positive cells in the population (*Figure 4a and d*). Altogether, this shows that an accumulation of extra-mitochondrial succinate upstream of heme synthesis in naive mESCs induces a 2C-like phenotype, reinforced by the inability of glycine to induce same effect (*Figure 4a*). In order to observe the levels of protein succinylation, and by extension the levels of succinate, in endogenous 2CLCs of the mESC population, we took advantage of a previously described reporter cell line for this 2C-state, characterized by the stable insertion of a construct containing a turboGFP-coding gene under the control of the MUERVL long terminal repeat (2 C:::turboGFP; *Ishiuchi et al., 2015*; *Rodriguez-Terrones et al., 2020*). The simultaneous observation of the endogenous GFP fluorescence, the absence of OCT4 (*Macfarlan et al., 2012*) and the immunostaining of the succinyl-lysine residues showed an increase in protein succinylation specifically in the 2C-like cells (GFP+; OCT4-) sub-population, as quantified by flow cytometry (*Figure 4e–f*).

Since we showed that the increase in the reprogramming of some mESCs to a 2C-like state after heme synthesis inhibition was the result of succinate exit from mitochondria, we then aimed to further confirm these results, by inducing a mitochondrial accumulation of succinate, independently of heme biosynthesis inhibition, using Atpenin A5 (AA5), an SDH inhibitor (*Miyadera et al., 2003*). AA5 induces the protein lysine succinylation (*Figure 5a*), an increase in the number of ZSCAN4-positive cells (*Figure 5b*) and the expression of 2 C markers (*Figure 5c and e*). These effects were counteracted by the addition of BM to prevent the succinate exit from the mitochondria (*Figure 5b*), confirming the involvement of succinate in the process.

## Succinate acts through the activation of its receptor, SUCNR1, to trigger the 2C-like program

Once in the cytosol, succinate could either inhibit the α-ketoglutarate (αKG)-dependent-dioxygenases or exit the cell through the monocarboxylate transporters (MCTs) and act as an extracellular signal molecule. Indeed, succinate is a reaction product of hydroxylation reactions catalyzed by αKG-dependent dioxygenases. Three families of dioxygenases are known to be sensitive to an increase in succinate concentration: the prolyl-hydroxylases (PDHs), the Ten-eleven translocation (TET)

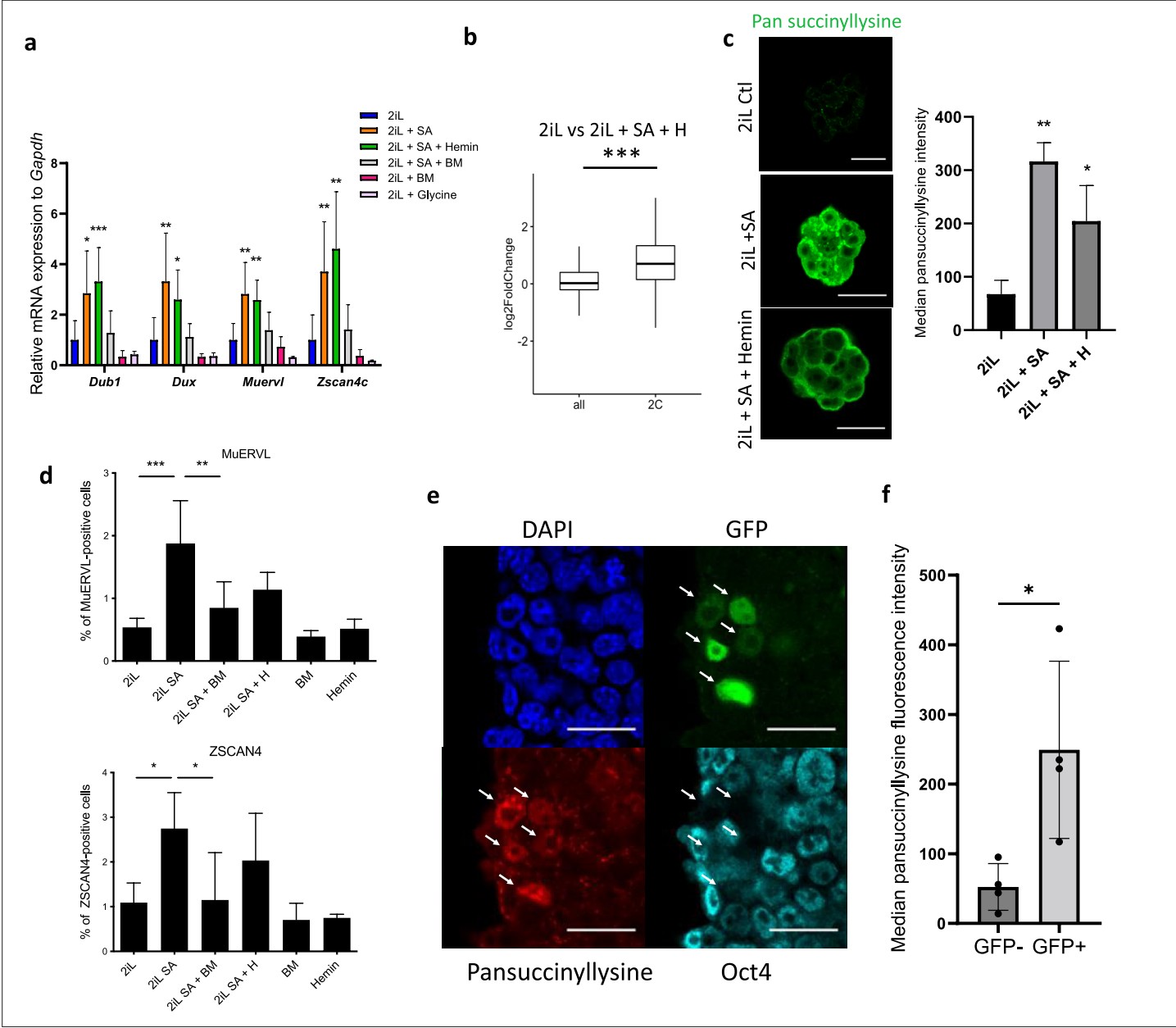

**Figure 4.** mESC '2C-like' reprogramming by SA is due to extramitochondrial succinate accumulation. (**a**) Relative expression of 2 C markers of mESCs assessed by RT-qPCR relative to *Gapdh* expression and normalized to 2iL naive control, in mESCs treated for 2 days with 0.5 mM SA (2iL SA), with or without 10 µM Hemin (2iL SA + H), 1 µM diethyl butylmalonate (2iL SA + BM), 1 µM BM alone (BM) or 10 mM glycine. S.D. * p<0.05, **p<0.01, ***p<0.001. ANOVA-1. n=3–6 independent biological replicates. (**b**) Boxplot of mean Log2FC of 2 C markers defined in *Macfarlan et al., 2012* or all analyzed mRNAs in 2iL + SA + H versus 2iL control cells. Statistical significance is calculated by Student T-test. p<0.001. (**c**) Immunostaining of succinylated lysine residues (green) in mESCs treated for 2 days with 0.5 mM SA, with or without 10 µM Hemin (SA + Hemin) and quantified by flow cytometry. Representative image of n=3 independent experiments. Scale bar = 20 µm. Results expressed as median fluorescence +/-S.D. * p<0.05, **p<0.01; ANOVA-1 (**d**) Percentage of MUERVL or ZSCAN4-positive cells in the whole population of naïve (2iL) mESCs or naive cells treated for 2 days with SA (2iL SA) with or without 10 µM hemin (2iL SA + H) or 1 µM diethyl butylmalonate (2iL SA + BM). n=4 independent biological replicates. Results expressed as mean +/-S.D. * p<0.05, **p<0.01, ***p<0.001; ANOVA-1. (**e**) Immunostaining of succinyllysine residues (Red) and Oct4 (cyan) of TBG4 cells (ES-E14TG2a mESCs with a 2C-GFP (green) reporter construct) (*Mills et al., 2018*). Representative image of n=3 independent experiments. Scale bar = 20 µm. Arrows indicate 2CLCs according to the GFP reporter. (**f**) Quantification of the median fluorescence intensity of the succinyllysine residues in the GFP +and GFP- populations of TBG4 mESCs separated by flow cytometry. n=4 independent biological replicates. * p<0.05, T-test.

The online version of this article includes the following figure supplement(s) for figure 4:

**Figure supplement 1.** Treatment with SA induces a delocalization of succinyllysine-modified proteins outside of mitochondria.

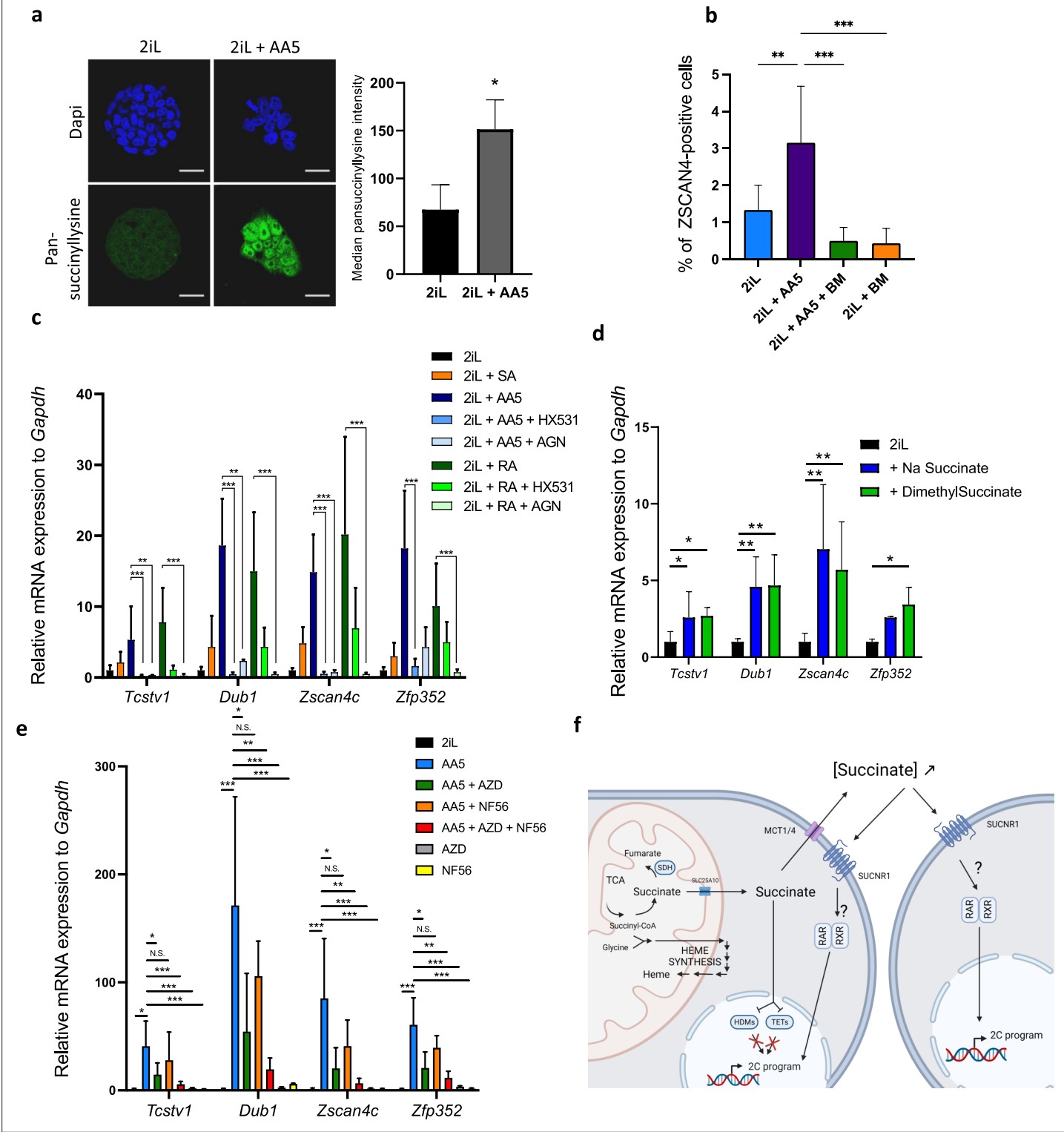

**Figure 5.** Inhibition of SDH leading to an increase in succinate accumulation recapitulates the increase in 2CLCs due to a leakage to the extracellular space. (**a**) Immunostaining of succinylated lysine residues (green) in mESCs in 2iL media or treated with 250 nM AA5. Representative image of n=3 independent experiments and quantified by flow cytometry. T-test. Scale bar = 20 μm (**b**) Percentage of ZSCAN4-positive cells in the whole population of naïve (2iL) mESCs or naïve treated with 250 nM AA5, with or without 1 μM diethyl butylmalonate (BM), counted from confocal micrographs with 10 images per conditions for at least 1000 cells per condition. n=3 independent biological replicates. ANOVA1 (**c**) Relative expression of the 2CLC genes in response to 250 nM AA5 or 0.5 μM retinoic acid (RA), with or without 10 μM HX531 and 100 nM AGN193109. (**d-e**) Relative expression of

*Figure 5 continued*

2C-gene markers of mESCs assessed by RT-qPCR relative to *Gapdh* expression in naïve 2iL mESCs and treated with 16 mM dimethyl succinate, 40 mM sodium succinate (Na succinate), 1 μM of MCT1 and 4 inhibitor (AZD3965; AZD) and 1 μM SUCNR1 receptor antagonist (NF-56-EJ40; NF56). Data shown as mean +/-S.D. *p<0.05, **p<0.01, ***p<0.001. ANOVA-1 followed by a Tukey post-test. n=4 independent biological replicates. (**f**) Graphical representation of the succinate-induced 2CLC reprogramming. Created with https://www.biorender.com/.

The online version of this article includes the following figure supplement(s) for figure 5:

**Figure supplement 1.** Treatment with SA or AA5 does not trigger HIF1α stabilization.

**Figure supplement 2.** Treatment with SA or AA5 triggers an increase of the methylation of histones and DNA that does not cause a 2C-like reprogramming.

methylcytosine dioxygenases and the histone demethylases (HDM). Of these three classes, we ruled out a role for PHDs, responsible for HIF1α degradation, as the abundance of this transcription factor was not increased by SA nor by AA5 treatment, demonstrating a lack of stabilization following a putative PHD inhibition (*Figure 5—figure supplement 1*). Since recent reports in the literature have highlighted a crucial role of the epigenetic landscape and a role of the TET proteins in the acquisition of the 2CLC phenotype (*Eckersley-Maslin et al., 2016*; *Huang et al., 2021*; *Schüle et al., 2019*; *Lu et al., 2014*; *Yang et al., 2016*), we then focused our attention on the histone 3 (H3) and DNA methylation landscapes and observed a robust increase in the amount of trimethylation of lysines 4, 9, and 27 of H3 (H3K4,-K9, -K27) along with an increase in the global abundance of 5mC when mESCs were treated with either SA or AA5 (*Figure 5—figure supplement 2a-d*). However, chemical inhibition of HDMs by JIB04 (*Wang et al., 2013*) and/or TET by TETin-C35 (*Singh et al., 2020*) does not trigger a rise in the proportion of 2CLCs or in 2 C marker expression (*Figure 5—figure supplement 2e-f*). We then suspected that succinate could act instead as a paracrine signal after its release in the extracellular media. Indeed, exogenous supplementation with a membrane impermeable (sodium succinate) or permeable (dimethylsuccinate) form of succinate were able to increase the 2CLC marker expression (*Figure 5d*). We further found that inhibition of the succinate exit through the plasma membrane, by targeting MCT1 and 4 with the AZD3965 inhibitor (*Beloueche-Babari et al., 2017*; *Reddy et al., 2020*) combined to inhibition of the succinate receptor SUCNR1 by NF-56-EJ40 (*Haffke et al., 2019*) is able to bring down the rise in 2CLC marker expression triggered by AA5 (*Figure 5e*). Recent reports have shown that retinoic acid (RA) also increases the 2CLC population, an effect mediated by the retinoic acid receptor RAR but not the RXRs (*Iturbide et al., 2021*). While the induction of the 2CLC gene signature in response to AA5 is similar to RA, the mode of action seems to involve both RAR and RXR since the effect is rescued by addition of both inhibitors, AGN193109 and HX531, respectively (*Figure 5c*). These findings strongly support succinate acting as a paracrine/autocrine signal upstream of RAR/RXR transcriptional activity. Together, these sets of data highlight the critical role of succinate in the acquisition and maintenance of pluri/totipotent states in mESCs by a totally new mechanism.

## Discussion

The comparison of two different genome-scale CRISPR screens in the exit of the naïve state of mouse and human ESCs (*Mathieu et al., 2019*; *Li et al., 2018*) revealed the importance of heme biosynthesis for the transition to another pluripotent state, the primed state. Using succinyl acetone (SA), a specific inhibitor of ALAD that catalyzes the second reaction of the pathway, and hemin to rescue the heme defect, we first confirmed the requirement of this pathway for the naive-to-primed transition in mESCs. This is in accordance with the embryonic lethality at the implantation stage of mouse embryos knockout for the heme synthesis pathway enzymes since the naive-to-primed transition recapitulates, in vitro, this critical step (shown for HMBS *Lindberg et al., 1996*, UROS *Bensidhoum, 1998*, UROD *Phillips et al., 2001*, CPOX *Conway et al., 2017* and FECH *Magness et al., 2002*). RNAseq analysis of naïve cells undergoing transition to the primed stage in the presence of SA with or without addition of hemin revealed a failure to properly activate the MAPK and TGFβ-SMAD pathways in response to heme biosynthesis inhibition. It has been previously demonstrated that the proper activation of these two pathways is required to proceed with the transition to the primed stage (*Tsanov et al., 2017*; *Jiapaer et al., 2018*; *Senft et al., 2018*; *Arman et al., 1998*). While the inhibition of SMAD3 was able to mimic the effect observed by heme synthesis inhibition, the inhibition of MAPK by MEK inhibition proved to be inefficient at blocking the process, in contradiction to its role in the

progression of pluripotency (*Tsanov et al., 2017*; *Jiapaer et al., 2018*; *Arman et al., 1998*). Such a connection between heme synthesis inhibition and signaling pathways has only been reported once in PC12 neuronal cells, with SA blocking the activation of the MAPK despite the presence of nerve growth factor (NGF) (*Zhu et al., 2002*). Thus, it seems to be a conserved mechanism among various cell identities/types, as it is also observed in our study for cells responding to FGF2 signals. While the link between heme and these crucial signaling pathways remains unknown, it brings to light a crucial importance of this metabolite and its synthesis pathway, so far poorly understood, in the context of pluripotency.

Aside from its effect on the exit of the naïve stage, we showed here that the inhibition of heme synthesis also triggers a reprogramming toward a 2C-like stage. Indeed, mESCs cultured in 2iL conditions have been previously defined as a heterogenous population that naturally includes a small percentage of cells displaying features of the two-cell stage embryo (*Macfarlan et al., 2012*; *Rodriguez-Terrones et al., 2020*). Interestingly, this 2CLC population is transient and cycles back and forth to a naive pluripotency state (*Macfarlan et al., 2012*; *Fu et al., 2020*). Unexpectedly, heme synthesis inhibition favors this reprogramming as shown by the increase in 2C-like markers in the whole population and the proportion of ZSCAN4 or MUERVL-positive cells. Strikingly, this effect is clearly dependent on the accumulation of succinate as it is (i) not rescued by hemin, (ii) blocked by inhibition of the mitochondrial succinate transporter and (iii) phenocopied by SDH inhibition. Additionally, MUERVL-positive cells spontaneously emerging in naïve ESC colonies endogenously display a high level of succinylated proteins, supporting a role for this metabolite in the identity of the 2C-like state. The role of several metabolites in the gain of 2CLC features has already been recently unveiled, highlighting a role for acetate, lactate, and D-ribose (*Rodriguez-Terrones et al., 2020*). This metabolite screening also showed a positive effect of succinate on the 2C-like features acquisition. Beyond its role in the cellular metabolism, we show here that the succinate accumulation outside mitochondria leads to a global reduction in cytosine demethylation activity of the TETs and histone demethylation as previously observed in cancers after accumulation of succinate or SDH mutations (*Letouzé et al., 2013*; *Xiao et al., 2012*). However, previous reports are somewhat discordant regarding the role of TETs in the acquisition of the 2CLC phenotype as their effect seems to be dependent on their interacting partners (*Eckersley-Maslin et al., 2016*; *Huang et al., 2021*; *Schüle et al., 2019*; *Lu et al., 2014*). For example, while TET2 could cooperate with PSPC1 (Paraspeckle Component 1) to reduce the expression of the retrotransposon *MuERVL* (*Guallar et al., 2018*), binding of the TET proteins with SMCHD1 (structural maintenance of chromosomes flexible hinge domain containing 1) prevents the demethylation of the *Dux* gene locus and thus prevents its expression (*Huang et al., 2021*). Succinate accumulation would result in a global decrease in the activity of all 3 TET isoforms, resembling those of a TET triple KO, already shown to induce the 2 C phenotype (*Lu et al., 2014*). The methylation landscape of histones, especially H3, is also highly dynamic both in vivo, at the time of the zygote genome activation (ZGA) that takes place at the 2C-stage, and in the 2CLC conversion with remodeling of H3K4me3, H3K9me3, and H3K27me3 (*Zhang et al., 2021*; *Wang et al., 2018*; *Yang et al., 2022*) (extensively reviewed in *Xia and Xie, 2020*). Similarly to the situation with the TETs, the action of HDM on the loss or acquisition of 2C-like features in vitro is complex, as loss of KDM1a (lysine demethylase 1 a) is shown to promote the expression of *Zscan4* and *MuERVL* (*Macfarlan et al., 2011*) whereas loss of KDM5a and b decreases the expression of the markers and blocks the ZGA in vivo (*Dahl et al., 2016*). However, while the literature highlights a role for these modifications of the epigenetic landscape in the 2CLC emergence, this is not supported by the data presented here. This discrepancy could be due to differences in the culture conditions, as these previous studies use mESC grown in serum +LIF conditions, while we use ground naïve mESCs (in 2iL and serum-free culture). Such difference has been previously described in the establishment of another pluripotent state of ESCs, the paused state (*Xu et al., 2022*). Interestingly, instead of an epigenetic rewiring as the major cause of 2CLC reprogramming, our data shed light on succinate acting as a paracrine/autocrine signal, able to trigger the emergence of 2CLCs by an MCT-dependent export and the activation of SUCNR1 expressing cells. Further studies are now needed to precisely dissect which downstream actors are truly responsible for the activation of the 2 C transcriptional program.

Our study thus brings an additional and different example of metabolic control of pluripotency, and adds succinate to previously reported metabolites such as S-adenosylmethionine (SAM) (*Sperber et al., 2015*), alpha-ketoglutarate (αKG) (*Carey et al., 2015*; *Tischler et al., 2019*), glutamine (*Tohyama*

*et al., 2016*), acetyl-CoA (*Moussaieff et al., 2015*), lactate or D-ribose (*Rodriguez-Terrones et al., 2020*) that are known to regulate the pluripotent states mostly through their contribution to modifications of the epigenetic landscape (reviewed in *Tsogtbaatar et al., 2020*). In addition to these previous studies, we highlight a role of succinate that goes beyond epigenetic landscape regulation but instead influences pluripotency regulation through protein succinylation and a paracrine effect. Further emphasis on the importance of succinate in early development is also demonstrated by the embryonic lethality of the SDH subunits in mice (*Piruat et al., 2004*; *Siebers et al., 2018*; *Lepoutre-Lussey et al., 2016*; *Takács-Vellai et al., 2021*). Altogether, these results indicate a critical role of both heme and succinate in the progression of the pluripotency continuum, ranging from the 2CLCs on one hand and to the acquisition of primed pluripotency on the other.

# Materials and methods
## Cell culture and ALAD ko generation

mESCs (ES-E14TG2a) or tbg4 mESCs (described in *Ishiuchi et al., 2015*) were cultured in N2B27 medium consisting of a 1:1 mixture of DMEM/F12 (Gibco, 31330–038) and Neurobasal Medium (Gibco, 21103–049) supplemented with 1 x N-2 Supplement (Gibco, 17502–048), 1 x B-27 Supplement (Gibco, 17504–044), 1/100 penicillin-streptomycin (Gibco, 15140–122), 1 x MEM nonessential amino acids (NEAA) (Gibco, 11140–035), 1 x GlutaMAX (Gibco, 35050–038), 1 x sodium-pyruvate (Gibco, 11360–039) and 0.1 mM β-mercaptoethanol (Gibco, 31350–010). Naïve mESCs were maintained on 0.2% gelatin (Sigma, G1393)-coated plates at a density of 50,000 cells/cm$^2$ and in N2B27 medium complemented with $10^3$ U/ml of mLIF (ESGRO, ESG1107), 3 µM of GSK3 inhibitor (CHIR99021) (Peprotech, 2520691) and 1 µM of MEK inhibitor (PD0325901) (referred to as 2iL) (SelleckChem, S1036). Cells were passaged every 2–3 days using accutase (Stemcell Technologies, #07920). Cells were then collected by centrifugation at 1200 rpm for 3 min and counted before seeding. The transition to EpiSC was obtained by transferring naïve mESCs on 15 µg/ml fibronectin (Gibco, 33010–018)-coated plates at a density of 30,000 cells/cm$^2$ and by supplementing the N2B27 medium with 12 ng/ml of bFGF (Peprotech, 100-18B) and 20 ng/ml of activin A (Peprotech, 120–14 P). Coating proteins were incubated 1 hr before seeding. mESCs were maintained at 37 °C, 5% CO$_2$ in a humidified incubator. Human ESCs, Elf1 line, were maintained of an irradiated MEF monolayer and grown in either RSeT (Stemcell Technologies) medium or in a medium composed of DMEM/F-12 media supplemented with 20% knockout serum replacer (KSR), 0.1 mM nonessential amino acids (NEAA), 1 mM sodium pyruvate, penicillin/streptomycin (all from Invitrogen), 0.1 mM β-mercaptoethanol (Sigma-Aldrich), 1 µM GSK3 inhibitor (CHIR99021), 1 µM of MEK inhibitor (PD0325901), 10 ng/ml human LIF (Chemicon), 5 ng/ml IGF1 (Peprotech) and 10 ng/ml bFGF. Cells were transferred to matrigel-coated plates prior to analysis. Exit from the naïve state triggered by growing the cells in mTeSR1 (StemCell technologies) for 4 days.

For knock-out generation, one million E14 mESC were electroporated with Cas9 (0.6 µM, Sigma) and gRNA (3 µM, Synthego) as RNP complex using Amaxa nucleofector (Human Stem Cell kit 2) in presence of 10 µM Hemin 10 uM. Individual colonies were hand-picked and plated into 96-well plates. DNA was extracted using Quick Extract DNA extraction solution (Epicentre#QE09050) and nested PCR was performed. The PCR product was purified using EXO-SAP enzyme (Thermo Fisher) and sent for Sanger sequencing analysis (through Genewiz).

Lines were routinely tested for mycoplasma and STR authenticated.

## mESC treatment

The heme inhibitor succinylacetone (SA) (Sigma, D1415) is used at a concentration of 0.5 mM. Hemin (Sigma, 51280) is used at a concentration of 10 µM in 0.1 N NaOH. Diethyl butylmalonate (BM) (Sigma, 112038) is used at a concentration of 1 mM. Atpenin A5 (Santa Cruz biotechnology, sc-202475) is used at 250 nM. SIS3 (Selleck chemicals, S7959) is used at 5 µM. NF-56-EJ40 (Axon Medchem 3056) is used at 1 µM. AZD3965 (Selleckchem S7339) is used at 1 µM. Exogenous succinate is provided as either 40 mM Sodium succinate (Sigma) or 16 mM dimethylsuccinate (Sigma) supplementation. HDM inhibition was achieved with 250 nM JIB04 (Medchem express HY-13953) and TET inhibition with 5 µM TETin-C35 (Aobious AOB11121). For retinoic acid experiments, RA, HY-14649, HX531 (HY-108521) and AGN193109 (HY-U00449) were used at 0.5 µM, 10 µM and 100 nM, respectively

(all from Medchemexpress). Inhibition of Sirt7 is achieved using 5 µM of the Sirt7 inhibitor 97491 (MedChemExpress).

## RNA extraction and RT-qPCR

RNA was extracted after 2 days of culture with the ReliaPrep RNA Tissue Miniprep System (Promega, Z6111) following manufacturer's protocol for non-fibrous tissue by adding RNA lysis buffer on pelleted cells. RNA concentrations were quantified with the Nanophotometer N60 (Implen). Reverse transcription (RT) was performed with the GoScript Reverse Transcriptase kit Random Primers (Promega, A2801) to convert 1 µg of RNA into cDNA. Briefly, RNA was mixed with RNAse-free water to obtain 1 µg of RNA in 12 µL and heated 5 min at 70 °C. Then, 8 µL of RT mix (4 µL random primers buffer, 2 µL enzyme, 2 µL RNAse free water) was added and the reaction was performed in a thermocycler (5 min at 25 °C, 60 min at 20 °C and 15 min at 70 °C).

The qPCR was performed on the ViiA 7 Real-Time PCR System (Thermo Fisher) with 10 ng of cDNA per reaction, SYBR Green GoTaq qPCR Master Mix (Promega, A6002) and primers listed in the *Table 2* at a final concentration of 300 nM. Altogether, 2 µL of cDNA (5 ng/µL), 1 µL of forward primer (6 µM), 1 µL of reverse primer (6 µM), 10 µL of Master Mix and 6 µL of RNAse-free water were added in each well. Relative expression was calculated using the $2^{-\Delta Ct}$ method with GAPDH as an endogenous control.

## Western blot analyses

Pellets of cells were lysed by adding protein lysis buffer 20 mM Tris-HCl; pH 7.5, 150 mM NaCl, 15% Glycerol, 2% SDS, 25 x protease inhibitor cocktail (PIC, cOmplete protease inhibitor cocktail, Roche 11697498001), 25 x phosphatase inhibitor buffer (PIB, composed of 25 mM $Na_3VO_3$, 250 mM 4-nitrophenylphosphate, 250 mM β-glycerophosphate and 125 mM NaF), 1% TRITON X-100, SuperNuclease (Sino Biologicals, 25 U/10 µL) and by pipetting up and down. The protein concentration was determined by Pierce protein assay (ThermoFisher, 22660). Samples were mixed with Laemmli buffer (SDS, β-mercaptoethanol, Bromophenol blue) and heated for 5 min at 95 °C before loading. An amount of 10 µg of proteins were loaded on SDS-containing 10 or 12% polyacrylamide gels. At the end of migration, proteins were transferred to PVDF membranes (IPFL00010) by liquid transfer. Membranes were then blocked in LI-COR Intercept blocking buffer PBS for 1 hr at RT and incubated overnight at 4 °C with the

**Table 2.** List of primers used in qPCR.

| Gene | Sequences (5' → 3') |
|---|---|
| Mouse | |
| DNMT3A | F: CTGCTGTGGAATACCCTGTTAG<br>R: CTTTCTACCTGCTGCCATACTC |
| ESRRB | F: GCACCTGGGCTCTAGTTGC<br>R: TACAGTCCTCGTAGCTCTTGC |
| FGF5 | F: GGGATTGTAGGAATACGAGGAGTT<br>R: CCAGAAGAATGGACGGTTGT |
| FGF15 | F: TGTTTCACCGCTCCTTCTTT<br>R: TTCTCCATCCTGTCGGAATC |
| GAPDH | F: CATGGCCTTCCGTGTTCCT<br>R: CCTGCTTCACCACCTTCTTG |
| KLF2 | F: CTAAAGGCGCATCTGCGTA<br>R: TAGTGGCGGGTAAGCTCGT |
| KLF4 | F: CCAGCAAGTCAGCTTGTGAA<br>R: GGGCATGTTCAAGTTGGATT |
| OCT4 | F: CACGAGTGGAAAGCAACTCA<br>R: AGATGGTGGTCTGGCTGAAC |
| OTX2 | F: TATCTAAAGCAACCGCCTTACG<br>R: AAGTCCATACCCGAAGTGGTC |
| REX1 | F: CCCTCGACAGACTGACCCTAA<br>R: TCGGGGCTAATCTCACTTTCAT |
| TFCP2L1 | F: GCTGGAGAATCGGAAGCTAGG<br>R: AAAACGACACGGATGATGCTC |
| ZIC2 | F: CAAGGTCCGGGTGCTTACC<br>R: ATTAAAGGGAGGCCCCGAATA |
| TBX3 | F: CTCCATTCCAGTTTGGTCAA<br>R: CAACAGCAGCCTGGTTACAC |
| OCT6 | F: TTTCTCAAGTGTCCCAAGCC<br>R: ACCACCTCCTTCTCCAGTTG |
| DNMT3B | F: GGCAAGGACGACGTTTTGTG<br>R: GTTGGACACGTCCGTGTAGTGAG |
| DUX | F: AAAGGAAGAGCATGTGCCAGC<br>R: GCAGTAAGCTGTCCTGGGAAC |
| ZFP352 | F: AAGTCCCACATCTGAAGAAACAC<br>R: GGGTATGAGGATTCACCCACA |
| TCSTV1 | F: TGAACCCTGATGCCTGCTAAGACT<br>R: AGATGGCTGCAAAGACACAACTGC |
| ZSCAN4C | F: CCGGAGAAAGCAGTGAGGTGGA<br>R: CGAAAATGCTAACAGTTGAT |
| MuERV-L | F: CCCATCATGAGCTGGGTACT<br>R: CGTGCAGATCCATCAGTAAA |
| DUB1 | F: GCAGGCCAACCTCAAACAG<br>R: CGCAGGGCTCTCCTAAATCTT |
| Human | |
| ZSCAN4 | F: TGGAAATCAAGTGGCAAAAA<br>R: CTGCATGTGGACGTGGAC |
| LEUTX | F: GCTACAATGGGGAAACTGG<br>R: CTCTTCCATTTGGCACGCTG |

*Table 2 continued on next page*

*Table 2 continued*

| Gene | Sequences (5' → 3') |
|------|---------------------|
| DUX4 | F: AGGAAGAATACCGGGCTCTG<br>R: AGTCTCTCACCGGGCCTAG |
| TFCP2L1 | Hs01011666_m1 |
| KLF4 | F: GGGAGAAGACACTGCGTCA<br>R: GGAAGCACTGGGGGAAGT |
| ESRRB | hs01584024_m1 |
| SALL1 | F: AGAGAACTCACACTGGAGAG<br>R: CATGTGTACCTTAAGATTGCCT |
| ETV4 | F: CGACTCTGAAGATCTCTTCC<br>R: TCATCACTGTCTGGTACCT |

primary antibodies. Membranes were washed three times for 5 min with PBS +0.1% Tween-20 (PBST) before and after 1 h-incubation with the secondary antibodies at RT. Detections were performed and quantified with Odyssey LI-COR scanner. Primary and secondary antibodies were both diluted in Licor PBST. GAPDH was used as loading control. Primary and secondary antibodies for GAPDH detection were incubated 30 min. Antibodies and dilutions used are: anti-GAPDH 1:20000(Sigma, G8795), anti-KLF4 1:200 (R&D, AF3158), anti-Oct4 1:100 (Santa Cruz, sc-5279), anti-OTX2 1:100 (R&D, AF1979), anti-pan-succinyllysines 1:300 (PTM-401), anti-TFE3 1:400 (Sigma, HPA023881), anti-ZSCAN4 1:400 (Millipore, AB4340), anti-MUERVL-GAG 1:200 (Novus, NBP2-66963), anti-ERK1/2 1:1000 (Cell Signaling 9102), anti-phosphoERK1/2 1:1000 (Cell Signaling 9101), anti-SMAD2/3 1:1000(Cell Signaling 8685), anti-SMAD3 1:1000 (Abcam ab52903).

## Immunofluorescence

Cells were seeded on coated glass cover (Assistent) slips 2 days before fixation with 4% paraformaldehyde (Sigma, 30525-89-4) for 15 min. Coating was performed for 1 hr with 15 µg/ml of fibronectin or with 3.5 µg/cm² Cell-Tak (VWR, 734–1081) diluted in 0.1 M sodium bicarbonate. Cells were permeabilized and blocked for 30 min incubation in blocking buffer (PBS, 0.1% TRITON, 1% BSA). Immunostaining was performed by an overnight incubation of cover slips at 4 °C on 30 µL drops containing primary antibody diluted in blocking buffer. After three washes of 5 min in blocking buffer, the cells were incubated for 1 hr in the dark and at RT with 30 µL drops containing secondary antibody and DAPI (Sigma, 10 236 276 001) diluted 1:1000 in blocking buffer. Cover slips were mounted with Mowiol after three washes of 5 min in blocking buffer. Analyses were performed with a Leica TCS SP5 confocal microscope (Leica microsystems). For 5mC analysis, cells were fixed for 15 min with 4% PFA, permeabilized for 10 min with PBS-0.5% Triton X-100 and treated with 4 N HCl for 20 min to denature DNA. Cells were then washed once with distilled water a 2 hr blocking with 2% BSA and 0.5% Triton X-100. After these steps, the staining process would resume as described above, using a 5mC primary antibody (Sigma SAB2702243) diluted 1:300 in blocking buffer (PBS, 0.1% TRITON, 1% BSA).

## Flow cytometry

Cells were detached by a 5 min incubation with Accutase and spun for 5 min at 400 *g*. The cell pellet was then washed with PBS-10 % FBS followed by another centrifugation. The pellet, containing at least 10⁶ cells, was fixed by adding 100 µl of 4% PFA for 15 min. After centrifugation, the cells were permeabilized with the Invitrogen permeabilization buffer (00833356) for 30 min at room temperature. Antibodies were incubated for 1h30 at room temperature in PBS-10 % FBS and briefly vortexed every 30 min. After three washes with PBS-10 % FBS, the cells were incubated with an anti-rabbit Alexa 647 1:1000 for 1 hr. Cell pellets were washed twice before being resuspended in PBS-10 % FBS and analysed with the FACSVerse machine. TBG4 ESCs (previously published *Eckersley-Maslin et al., 2016*) were used for pan-succinyllysine levels measure in 2CLCs, by gating the turboGFP +population and measuring the alexa 647 intensity.

## RNA sequencing

Sequence libraries were prepared with the Lexogen QuantSeq 3' mRNA-Seq library prep kit according to the manufacturer's protocol. Samples were indexed to allow for multiplexing. Library quality and size range were assessed using a Bioanalyzer (Agilent Technologies) with the DNA 1000 kit (Agilent

Technologies, California, USA). Libraries were subsequently sequenced on an Illumina HiSeq4000 instrument. Single-end reads of 50 bp length were produced with a minimum of 1 M reads per sample.

Quality control of raw reads was performed with FastQC v0.11.7, available online at: http://www.bioinformatics.babraham.ac.uk/projects/fastqc. Adapters were filtered with ea-utils fastq-mcf v1.05 (Erik Aronesty (2011), ea-utils: "Command-line tools for processing biological sequencing data; https://github.com/ExpressionAnalysis/ea-utils copy archived at *Expression Analysis, 2021*). Splice-aware alignment was performed with HiSAT2 against the mouse reference genome mm10. Reads mapping to multiple loci in the reference genome were discarded. Resulting BAM files were handled with Samtools v1.5 (*Li et al., 2009*). Quantification of reads per gene was performed with HT-seq Count v2.7.14. Count-based differential expression analysis was done with R-based Bioconductor package DESeq2. Reported p-values were adjusted for multiple testing with the Benjamini-Hochberg procedure, which controls false discovery rate (FDR).

## Data analysis

TMM normalized rLog transformed counts were used for Principal Component analysis using R package PCATools. Gene set enrichment analysis (GSEA) was made on gene list ranked on Log2FC using R package ClusterProfiler (*Yu et al., 2012*). For genes with FC >2 in MUERVL::Tomato$^+$ list from *Macfarlan et al., 2012*, Z score was calculated from TMM-rLog transformed counts and plotted as heatmap using R package Heatmap.plus. Analysis was made using statistical programming language R.

## Statistical analysis

One-way ANOVA statistical test followed Turkey multiple comparison test or student T-Tests were conducted on all results when indicated using GraphPad Prism version 9.1.1, GraphPad Software, San Diego, California USA, https://www.graphpad.com.

## Acknowledgements

We are grateful to Dr. Maria Helena Padilla-Torres (Institute of epigenetics and stem cells; Helmholtz Zentrum München) for kindly providing the 2 C:::turboGFP mESC cell line. We also thank the Morphym platform – UNamur for the help with confocal and flow cytometry analysis.

This work was supported by the Fonds de la Recherche Scientifique – FNRS, 5, rue d'Egmont, 1000 Brussels. DD and MC are recipient of a (Fonds de la Recherche dans l'Industrie et l'Agriculture [FRIA], Belgium), fellowship and S.M. is recipient of a (Fonds National de la Recherche Scientifique [FNRS], Belgium) fellowship.

## Additional information

### Funding

| Funder | Grant reference number | Author |
|---|---|---|
| Fonds De La Recherche Scientifique - FNRS | | Sébastien Meurant |
| Fonds de la Recherche dans l'Industrie et l'Agriculture | | Damien Detraux Marino Caruso |

The funders had no role in study design, data collection and interpretation, or the decision to submit the work for publication.

### Author contributions

Damien Detraux, Conceptualization, Data curation, Formal analysis, Funding acquisition, Validation, Investigation, Visualization, Methodology, Writing – original draft, Writing – review and editing; Marino Caruso, Software, Investigation, Visualization, Methodology, Writing – review and editing; Louise Feller, Investigation, Methodology, Writing – review and editing; Maude Fransolet, Investigation;

Sébastien Meurant, Investigation, Writing – review and editing; Julie Mathieu, Conceptualization, Supervision, Writing – review and editing; Thierry Arnould, Supervision, Writing – review and editing; Patricia Renard, Conceptualization, Supervision, Writing – original draft, Writing – review and editing

**Author ORCIDs**
Damien Detraux (iD) http://orcid.org/0000-0002-9704-2076
Sébastien Meurant (iD) http://orcid.org/0000-0003-0711-9605
Patricia Renard (iD) http://orcid.org/0000-0003-4144-3353

**Decision letter and Author response**
Decision letter https://doi.org/10.7554/eLife.78546.sa1
Author response https://doi.org/10.7554/eLife.78546.sa2

---

## Additional files

**Supplementary files**
• MDAR checklist

**Data availability**
Sequencing data have been deposited in GEO under the accession GSE178089. Data files for the western blot images have been added as raw images of scans.

The following dataset was generated:

| Author(s) | Year | Dataset title | Dataset URL | Database and Identifier |
|---|---|---|---|---|
| Detraux D, Caruso M | 2021 | The critical role of heme synthesis in the regulation of pluripotent states is succinate-dependent | https://www.ncbi.nlm.nih.gov/geo/query/acc.cgi?acc=GSE178089 | NCBI Gene Expression Omnibus, GSE178089 |

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
