## [Editor Report]

In their study, Detraux D and colleagues provide compelling evidence demonstrating a role for heme biosynthesis on FGF-ERK and TGF β signalling and exit from naïve pluripotency, and in controlling the 2-cell-like cell state. The observations and conclusions provided by the authors are convincing and potentially relevant in the field of pluripotent cell state transitions.

---

## [Decision Letter]

**Decision letter after peer review:**

Thank you for submitting your article "A critical role for heme synthesis and succinate in the regulation of pluripotent states transitions" for consideration by *eLife*. Your article has been reviewed by 2 peer reviewers, and the evaluation has been overseen by a Reviewing Editor and Mone Zaidi as the Senior Editor. The reviewers have opted to remain anonymous.

Essential revisions:

1) Are the levels of heme itself, or the mentioned 7 enzymes of the heme pathway regulated during differentiation?

2) Further experiments are needed to determine whether the lack of proper Tgf β and FGF-ERK signalling activation are cause or consequence of the observed differentiation defects. How do cells sense heme deficiency and does how heme deficiency result in inability to activate ERK/MAPK and TGF β signaling?

3) Can the authors attempt to titrate pathway inhibition to a similar level as observed in heme pathway deficient ESCs? Furthermore, can the differentiation defect be rescued upon overstimulation of FGF-ERK and TGF β to reach WT levels?

4) Analyse cell proliferation and viability after SA (or AA5) treatment. If there is an impact on cellular health, this needs to be reported and taken into careful consideration when interpreting results.

5) The statement made in lines 150-152 and shown as Suppl. Figure 2b should be further supported by proper quantification using replicate assays.

6) Expression levels in TS cells or in trophoblast tissue must be used as control to judge the effect size. The statement that SA treatment expands the lineage potential of ESCs needs to be supported by appropriate and statistically strong data beyond the increase in some marker genes (which are also expressed in embryonic tissues).

7) Please provide direct evidence for disruption of heme biosynthesis directly instructing cells to take on a 2CLC state. How strong is the effect of inhibiting heme synthesis and succinate in inducing the 2CLC population as the absolute number of Zscan4+ or MuERVL+ cell is < 3%? The authors could use other 2CLC inducers (such as acetate (Rodriguez-Terrones, 2020) and RA (Iturbide, A., 2021) as a control, for comparison. The functional assay of 2 cell-like cells after heme synthesis inhibition should be test in vivo by 8-cell or blastocyst injection.

8) Quantification and comparison of pan succinate levels in Figures4c and 5a would be required for interpretation.

9) What happens with positive enriched genes in the SA-treated condition? Upregulation plot should be added to Figure 2a and discussed.

10) In the section describing the upregulation of 2-cell-stage-related genes when SA is added, please add a scramble as control, treat 2iL cells with an inhibitor of a different metabolic pathway for 48h and analyze by rt-PCR a) the 2-cell-related genes, b) the differentiation towards trophoblast cells.

11) The authors provide a very detailed molecular analysis on the effect of heme synthesis inhibition on the pluripotency exit. However, functional characterization (such as teratoma formation) is important to tell whether these naive mESC after treatment with heme synthesis blocker have any defects on the pluripotency exit and differentiation.

12) Although it is clear that heme synthesis is important for the mouse pluripotency status transition, it is not clear whether this is a mechanism present in human as well. The authors presented bioinformatic data suggesting that this might be the case. Does heme synthesis blockade increase the emergence of human 2 cell-like cells?

13) A genetic approach would complement and strengthen the specificity of the conclusions obtained with chemicals. For example, the inhibition of heme synthesis by SA should also be confirmed by knockout or knockdown of proteins in this pathway like ALAD, PBGD et al.

*Reviewer #1 (Recommendations for the authors):*

The authors will find below a list of my concerns.

1. For readers of the stem cell field a more detailed introduction into the heme synthesis pathway needs to be included in the introduction,

2. The statement made in lines 150-152 and shown as Suppl. Figure 2b should be further supported by proper quantification using replicate assays.

3. Lines 207-209. The rationale and the conclusion behind the experiment shown in Suppl. Figure 4 must be better explained.

4. Line 98: the word 'prevented' is too strong for the observed effect. Oct6, Otx2 and Dnmt3b are clearly upregulated, merely less than in WT.

5. The authors could put their findings in context of data on the impact of modulating the intracellular αKG/succinate ratio on pluripotency maintenance (https://doi.org/10.1038/nature13981; https://doi.org/10.15252/embj.201899518).

6. Does BM also rescue the exit from pluripotency phenotype?

7. Quantification and comparison of pan succinate levels in Figures4c and 5a would be helpful and required for interpretation. As there is basically no staining detectable in untreated cells in 4a and some in 5a, it is unclear whether SA and AA5 effects are similar in size. It appears that AA5 treatment has a much weaker effect.

*Reviewer #2 (Recommendations for the authors):*

– In the first part, regarding the analysis of previously published results of CRISPR-Cas9 screens (lines 77-87): It would be ideal to use an alternative annotation tool for gene ontology analyses. DAVID is already outdated. If the finding is corroborated with an alternative method, it not only gives strength to the initial observations but allows finding new targets of interest.

– The second title, "Heme synthesis inhibition prevents the activation of key developmental signaling pathways," needs to be more specific. Because heme is a critical component of general cell homeostasis, it is evident that the inhibition will affect key signaling pathways, not only in development but in any stage of mammal's life.

In this same section, what happens with positive enriched genes in the SA-treated condition? It should be added the up-regulation plot side Figure 2a, and a correspondent discussion about it.

– In the section describing the upregulation of 2-cell-stage-related genes when SA is added, it is needed to add a scramble-inhibition as control. Treat 2iL cells with an inhibitor of a different metabolic pathway for 48h and analyze by rt-PCR (a) the 2-cell-related genes, (b) the differentiation towards trophoblast cells.

– What happens with the glycine levels after heme synthesis inhibition? Does glycine play a role in the induction of a 2-cell-like state? Ideally authors should include an experiment to test this role.

– The following is optional, but it could be a considerable improvement to provide the bulk RNA seq comparing the inhibition of SUCNR1 vs. the activation induced by succinate, to see all other pathways that could crosstalk and describe better the mechanism of the succinate in pluripotent/totipotent states.

– The authors provide a very detailed molecular analysis on the effect of heme synthesis inhibition on the pluripotency exit. However, functional characterization (such as teratoma formation) is important to tell whether these naive mESC after treatment with heme synthesis blocker have any defects on the pluripotency exit and differentiation.

– Although it is clear that heme synthesis is important for the mouse pluripotency status transition, it is not clear whether this is a mechanism present in human as well. The authors presented bioinformatic data suggesting that this might be the case. It is important that the authors perform functional studies in the human cells to further substantiate this point.

– It is not clear how the cell senses heme deficiency and how heme deficiency results in inability to activate ERK/MAPK and TGF β signaling. Could the authors provide comments into this aspect?

– How strong the effect of inhibiting heme synthesis and succinate in inducing the 2CLC population are as the absolute number of Zscan4+ or MuERVL+ cell are quite low (< 3%)? I suggest that the authors include other 2CLC inducers such as acetate (Rodriguez-Terrones, 2020) and RA (Iturbide, A., 2021) as a control for comparison.

– The authors performed the 2CLC experiment with 2iL condition. However, a quick review of the literature indicates that many of these studies have been done in FBS/LIF system. Can the authors comment on the difference between 2iL condition and FBS/LIF in the study of 2CLC?

– The authors analyzed that heme synthesis is also important human pluripotent state transition. It will be interesting to see the scenario in human pluripotent stem cells, which will largely make the study difference, especially if the heme synthesis blockade will also increase the emergence of human 2 cell-like cells.

– The functional assay of 2 cell-like cells after heme synthesis inhibition should be tested in vivo by 8-cell or blastocyst injection

– The whole study of inhibition is chemical achieved. As a supplement, the authors should also test with genetic intervention instead of sole chemicals. For example, the inhibition of heme synthesis by SA should also be confirmed by knockout or knockdown of proteins in this pathway like ALAD, PBGD et al. Also similar with AA5.

– To make the finding here more convincing and significant, it will be good to confirm in embryos.

– The manuscript should include the metabolism part in the introduction instead of using large paragraphs in the main text to introduce these pathways and previous studies.

---

## [Author Response]

Essential revisions:1) Are the levels of heme itself, or the mentioned 7 enzymes of the heme pathway regulated during differentiation?

We thank the reviewer to point that it is important to verify whether there could be a regulation of heme biosynthesis during the naïve-to-primed transition. To answer this, we explored the expression of the 8 genes encoding enzymes in the pathway, from this study and previously published ones, both in human and mouse and from in vivo blastocysts or in vitro models (Grow *et al.,* 2015; Sperber *et al.,* 2015; Di Stefano *et al.,* 2018; Nakamura *et al.,* 2016). While some of the enzymes show a slight change during the exit of the naïve state, none are consistent between the different studies and no significant trend for the pathway is observed. This has been added as Table 2.

2) Further experiments are needed to determine whether the lack of proper Tgf β and FGF-ERK signalling activation are cause or consequence of the observed differentiation defects. How do cells sense heme deficiency and does how heme deficiency result in inability to activate ERK/MAPK and TGF β signaling?3) Can the authors attempt to titrate pathway inhibition to a similar level as observed in heme pathway deficient ESCs? Furthermore, can the differentiation defect be rescued upon overstimulation of FGF-ERK and TGF β to reach WT levels?

We thank the reviewers for their highly relevant comments. We have addressed these two questions simultaneously.

Heme deficiency is canonically sensed in mammals through the nuclear factor BACH1 (reviewed in (Zhang *et al.,* 2018)) or the activation of the Integrated Stress Response (ISR) through the Heme Responsive Inhibitor (HRI) or EIF2AK1 (reviewed in (Pakos‐Zebrucka *et al.,* 2016)).

BACH1 heterodimerized with MAF proteins (among others) binds to DNA and represses the transcription of target genes. Upon heme binding to BACH1, the complex dissociates and BACH1 is exported in the cytosol, thus relieving the repression of target genes. As seen in these immunofluorescence analyses of BACH1 subcellular localization (Figure 2 – Supplement Figure 1a), the abundance of nuclear BACH1 is comparable in 2iL, EPI, and EPI +SA conditions, indicating that the inhibition of heme synthesis does not affect BACH1 activity. On the opposite and as expected, the addition of hemin provokes the nuclear exclusion of BACH1. This data indicates that BACH1 is not responsible for the heme deficiency sensing in our model. This was added to the manuscript.

As another putative way of heme depletion to act on the transition from naïve to primed, we also investigated the role of the ISR. Indeed, various environmental and pathological conditions, including protein homeostasis (proteostasis) defects, nutrient deprivation, viral infection, and oxidative stress, activate the ISR to restore the balance by reprogramming gene expression, all converging to the phosphorylation and activation of the eukaryotic translation initiation factor eIF2. Heme depletion is known to be one of the stresses activating this pathway, allowing the phosphorylation and activation of the heme-regulated inhibitor (HRI) kinase, in turn phosphorylating EIF2α. To assess the activation of this pathway, we targeted two levels: the phosphorylation of EIF2α by western blot analysis and the global protein synthesis levels with a SUnSET assay (Schmidt *et al.,* 2009) (Figure 2 – Supplement Figure 1b-c). The increase in EIF2α phosphorylation (Figure 2 – Supplement Figure 1b), a direct result of the activity of HRI in absence of heme, indicates the activation of a kinase upstream of the pathway. The global reduction in protein synthesis, observed with the levels of puromycin-labelled peptides (Figure 2 – Supplement Figure 1c), further confirms the activation of the ISR upon inhibition of heme synthesis.

To further explore the possible implication of this ISR-HRI axis in the inhibition of native-to-primed transition induced by heme synthesis inhibition, we used a chemical activator of HRI (BTdCPU; (Chen *et al.,* 2011)) to assess whether it could induce a similar effect as SA or not. Treatment of cells with 2µM BTdCPU, reduces protein synthesis to a similar extent that SA (Figure 2 – Supplement Figure 1d), but fails at preventing the naïve stage exit (Figure 2 – Supplement Figure 1e), indicating an HRI-independent mechanism. These results were added to the manuscript and are now part of Figure 2 – Supplement Figure 1.

In a final attempt to identify the mechanism behind the failure of MAPK and TGFβ activation following heme synthesis inhibition, we also searched the GO terms associated with heme binding that could regulate the pathways or receptor mentioned. The analysis did not reveal any putative regulators. However, following an interesting suggestion from the reviewers, we observed that an increase in Activin A or FGF2 concentration (2- or 3-fold increase) did not rescue the naïve exit defect (Figure 2 – Supplement Figure 1j), suggesting a strong intracellular mechanism. Since activin and FGF are the sole signals used to trigger the exit, an impairment of those would indeed prevent it, especially for TGFβ pathway, as shown in the supplementary figure 3i. Thus, instead of chemically block the TGFb pathway, we decided to investigate the temporal activation of SMAD3 phosphorylation during the exit of naïve stage, with or without SA (Figure 2b-c). Quantification of the phosphorylation kinetics of SMAD3 indicates a significant decrease starting after 24h post induction of the exit when treated with SA (Figure 2b-c). This correlates with the gene expression kinetics of Fgf5 and Tbx3 (Figure 2d), suggesting a key role of the SMAD3 pathway perturbation in the effect.

Mechanistically, it is known that heme synthesis inhibition raises the levels of succinyl-CoA and succinate, since 8 moles of succinyl-CoA and 8 moles of glycine are needed to produce 1 mole of heme, heme synthesis acting thus as a “succinyl-CoA sink” (Atamna 2004). In addition, succinate accumulation upon heme synthesis inhibition is reinforced by the drastically reduced abundance of succinate dehydrogenase (SDH) consuming succinate in the TCA that consumes succinate. Indeed, SDH is a heme-dependent complex known to be destabilized by the loss of heme nicely reviewed in (Kim *et al.,* 2012), as confirmed in Figure 2e, showing the drastically reduced SDH abundance after heme synthesis inhibition. Together this indicates a possible accumulation of succinate in response to heme synthesis inhibition, an accumulation that could cause the transition defect. To test this hypothesis, we used butylmalonate as an inhibitor of the dicarboxylate carrier (DIC; SLC25A10) to prevent leakage of succinate from the mitochondrial matrix to the cytosol, and this nicely resumed the transition as shown by the gene expression profile of mESCs in the EpiLC + SA + BM condition (Figure 2f). Succinate accumulation and/or SDH ablation is known to increase protein lysine succinylation (Smestad et al., 2018), an effect that is observed upon SA treatment (Figure 2g) and that we mimicked by inhibiting Sirt7, a cytosolic and nuclear desuccinylase (Figure 2g). The ability of this molecule to mimic SA in the pathway inhibition (Figure 2h-i) and to prevent the exit of the naïve state (Figure 2f) points at succinylation events as the cause of the transition defects.

This new data is now inserted, commented and discussed in the manuscript as part of figure 2 or Figure 2 – Supplement Figure 1.

4) Analyse cell proliferation and viability after SA (or AA5) treatment. If there is an impact on cellular health, this needs to be reported and taken into careful consideration when interpreting results.

We used a live:dead staining kit (Invitrogen L3224, using calcein-AM and ethidium homodimer-1) to analyze the survival fraction of cells after two days in culture, following either SA or AA5 treatment. No significant change was observed (Author response image 1). The cell cycle analysis was performed by flow cytometry with propidium iodide. Overall, SA treatment increases slightly the G2 cell population, while AA5 increases the S population (Author response image 1). Since these two drugs induced similar effects in the acquisition of 2CLCs in the naïve population, it is thus unlikely that the effects of these molecules on cell viability and proliferation do affect the results obtained and presented.

**Author response image 1. sa2fig1:** Effect of heme synthesis (SA) and SDH (AA5) inhibitors on mESC homeostasis. (A) Cell survival analysis of mESCs in 2iL naïve control, in mESCs treated with 0.5 mM SA or 250 nM Atpenin A5 (AA5) for 2 days, assessed using calcein-AM and ethidium homodimer-1 (Invitrogen) in confocal microscopy, expressed as % of Calcein-AM-positive cells to the whole population and represented as means +/- S.D. N.S p>0.05, ANOVA1. n=3 biological independent replicates. (B) Cells cycle analysis of mESCs in 2iL naïve control, in mESCs treated with 0.5 mM SA or 250 nM Atpenin A5 (AA5) for 2 days and assessed using propidium iodide by flow cytometry and quantified with FLowJo software. n=3 biological independent replicates.

5) The statement made in lines 150-152 and shown as Suppl. Figure 2b should be further supported by proper quantification using replicate assays.6) Expression levels in TS cells or in trophoblast tissue must be used as control to judge the effect size. The statement that SA treatment expands the lineage potential of ESCs needs to be supported by appropriate and statistically strong data beyond the increase in some marker genes (which are also expressed in embryonic tissues).

Questions 5 and 6, both related to the increase in cell potential, are thus treated simultaneously. While further quantification of gene expression (*Cdx2, Tead4, Gata3, Eomes, Hand1, Elf5, Krt18, Cited1, Tfap2c and Ets1*) following SA or AA5 treatment still shows a trend toward an increase of differentiation efficiency (Author response image 2), we observed high variability and the increase in the expression of trophoblast-specific genes was not significant (n=6). While this still could be interesting, we hypothesize that since the number of 2CLCs is increased from 1 to 3 % upon SA or AA5 treatment, the effect on lineage potential will remain limited (see also the response to question 7). Because of the lack of significance, we removed the data and related discussion on the extended potential from the manuscript.

**Author response image 2. sa2fig2:** Expression of trophoblast lineage markers after differentiation. Relative mRNA expression of trophoblasts lineage markers of mESCs differentiated for 6 days after culture for 2 days in naïve 2iL control condition (blue) or with 0.5 mM SA (orange), or 250 nM AA5 (grey), assessed by RT-qPCR. Data shown as means +/- S.D. n=6 biological independent replicates.

7) Please provide direct evidence for disruption of heme biosynthesis directly instructing cells to take on a 2CLC state. How strong is the effect of inhibiting heme synthesis and succinate in inducing the 2CLC population as the absolute number of Zscan4+ or MuERVL+ cell is < 3%? The authors could use other 2CLC inducers (such as acetate (Rodriguez-Terrones, 2020) and RA (Iturbide, A., 2021) as a control, for comparison. The functional assay of 2 cell-like cells after heme synthesis inhibition should be test in vivo by 8-cell or blastocyst injection.

The updated figure 5c now includes RA as a positive control as presented in Iturbide et al. 2021 and as suggested by the reviewers. Interestingly, we observe that the gene expression levels of the 2CLC markers after RA treatment increase to a similar extent than with AA5, but not SA.

These authors showed that the RA-induced 2CLC increase is due to activation of the RARγ since inhibition of RXR (with the HX531 inhibitor) does not rescue the effect while a RAR inhibitor (AGN193109) does. Very interestingly, the AA5-induced 2CLC signature seems to depend on both receptors as HX531 (RXR inhibitor) and AGN193109 (pan-RAR inhibitor) do inhibit the AA5-induced 2CLC gene expression signature.

Since genetic ablation of heme synthesis enzymes is embryonic lethal around the implantation time (Lindberg *et al.,* 1996; Bensidhoum *et al.,* 1998; Phillips *et al.,* 2001; Conway *et al.,* 2017; Magness *et al.,* 2002), we expect this experiment to bear toxicity for the embryo, especially since accumulation of heme synthesis intermediate is known for its toxicity (Lämsä *et al.,* 2012; Handschin *et al.,* 2005). Furthermore, injection of the 2CLC in early embryos is rarely used, including in Iturbide, 2021 or Rodriguez-Terrones, 2020.

8) Quantification and comparison of pan succinate levels in Figures4c and 5a would be required for interpretation.

We thank the reviewers for pointing at this issue. We performed a quantification of the pan succinyllysine residue levels by flow cytometry and the results are now shown in addition to the more qualitative confocal micrographs in figures 4 and 5. These results better support our claims and conclusion.

9) What happens with positive enriched genes in the SA-treated condition? Upregulation plot should be added to Figure 2a and discussed.

The upregulation plot has been added to figure 2a. A majority of the Gene ontology (GO) terms found upregulated in EpiLC SA vs EpiLC includes cytochrome-associated proteins for xenobiotic detoxification or oxidative phosphorylation. This probably corresponds to a compensatory mechanism since the absence of heme reduces the abundance of proteins encoded by to those genes, as previously shown (Lämsä *et al.,* 2012; Vinchi *et al.,* 2014).

10) In the section describing the upregulation of 2-cell-stage-related genes when SA is added, please add a scramble as control, treat 2iL cells with an inhibitor of a different metabolic pathway for 48h and analyze by rt-PCR a) the 2-cell-related genes, b) the differentiation towards trophoblast cells.

To validate the metabolic effect of SA or AA5 on the 2CLC population, we thank the reviewers for suggesting using alternative metabolism inhibitors. We used two different metabolic inhibitors: Etomoxir as an inhibitor of Carnitine Palmitoyl Transferase (CPT-1), thus blocking fatty acid oxidation, and 6-aminonicotinamide (6-AN), an antimetabolite blocking the pentose phosphate pathway (PPP) metabolism. These two molecules fail to induce a 2CLC signature (Author response image 3).

**Author response image 3. sa2fig3:** Inhibitors of pentose phosphate pathway and fatty acid oxidation do not trigger a 2CLC signature. Relative expression of 2C markers of mESCs assessed by RT-qPCR relative to Gapdh expression and normalized to expression level of 2iL naïve control, in mESCs treated with 50 µM Etomoxir or 50 µM 6-aminonicotinamide for 48 h. n=3 independent biological replicates.

11) The authors provide a very detailed molecular analysis on the effect of heme synthesis inhibition on the pluripotency exit. However, functional characterization (such as teratoma formation) is important to tell whether these naive mESC after treatment with heme synthesis blocker have any defects on the pluripotency exit and differentiation.

As explained in the Discussion section, the fact that heme synthesis knock-out is embryonic lethal around the implantation stage in mouse (Lindberg *et al.,* 1996; Bensidhoum *et al.,* 1998; Phillips *et al.,* 2001; Conway *et al.,* 2017; Magness *et al.,* 2002) strongly suggests an implantation defect, hence, pluripotency exit. This is further supported by the loss of Oct4 in ALAD KO mESCs after removal of hemin (Supp. Figure 2b). Furthermore, teratoma formation assay or embryoid body formation assays after a temporary blockade (2 days) of heme synthesis through SA would quickly be attenuated as heme synthesis would resume after injection, especially since these assays extends for weeks.

12) Although it is clear that heme synthesis is important for the mouse pluripotency status transition, it is not clear whether this is a mechanism present in human as well. The authors presented bioinformatic data suggesting that this might be the case. Does heme synthesis blockade increase the emergence of human 2 cell-like cells?

To answer this question, we used Elf1 hESC cells (Ware *et al.,* 2014) grown in RSeT media to mimic the naïve conditions, and triggered the exit toward the primed stage for 4 days in TeSR media (both commercially available, StemCell technologies), with or without 0.5 mM of SA. We observe a significant prevention of the loss of naïve markers, as in mESCs, while not preventing the increase in a primed signature. This has been updated and commented in the manuscript, with an updated Figure 1 – Supplement Figure 1c.

In addition, naïve Elf1 hESCs maintained either in RSeT medium or in 2iL-I-F (GSK3 and MEK inhibitors, LIF, IGF1 and FGF2) media (Sperber *et al.,* 2015) treated with 0.5 mM of SA for 2 days, show a significant upregulation of markers associated with the 8-cell stage (the human homolog of the 2CLC), corresponding to the zygote genome activation event (Taubenschmid-Stowers *et al.,* 2022), suggesting a similar mechanism (Figure 3g, Author response image 4).

**Author response image 4. sa2fig4:** Heme synthesis inhibition in naïve hESCs triggers a ZGA gene signature. Relative expression of ZGA related genes in hESCs assessed by RT-qPCR relative to TBP expression and normalized to naïve control, with or without 0.5 mM of SA. n=3 independent biological replicates. Results expressed as means +/- S.D. * p < 0.05, **p < 0.01, ***p < 0.001 ; t-tests.

13) A genetic approach would complement and strengthen the specificity of the conclusions obtained with chemicals. For example, the inhibition of heme synthesis by SA should also be confirmed by knockout or knockdown of proteins in this pathway like ALAD, PBGD et al.

We thank the reviewer for this important comment. Using the CRISPR-Cas9 technology, we generated an ALAD KO E14-mESC line using CRISPR/Cas9 (Figure 1 – Supplement Figure 1a). To maintain a healthy population, this line is grown in a medium supplemented with hemin and removal of this supplement allows to reveal the heme synthesis inhibition. As shown in the updated figure 3 (Figure 3d), removal of hemin in the culture medium for 2 days induces a strong increase in 2CLC marker expression.

We also performed the transition from naïve to primed in these ALAD KO mESCs. Genetic ablation of ALAD also prevents mESCs to properly exit the naïve state as the naïve marker expression is maintained. On the other hand, primed markers, except Dnmt3b, are upregulated (Figure 1 – Supplement Figure 1b). While this is in apparent contrast to the chemical inhibition of ALAD, the complete loss of ALAD seems to perturb the pluripotency state as seen by the complete loss of Oct4 expression, thus impairing the proper transition. This loss of Oct4 transcript could be due to the complete loss of heme, impairing the formation of G-quadruplexes, known to promote Oct4 gene expression (Renčiuk *et al.,* 2017; Gray *et al.,* 2019).

References

Bensidhoum M, Larou M, Lemeur M, Dierich A, Costet P, Raymond S, Daniel JY, De Verneuil H and Ged C (1998) The disruption of mouse uroporphyrinogen III synthase (uros) gene is fully lethal. *Transgenics* 2: 275–280

Chen T, Ozel D, Qiao Y, Harbinski F, Chen L, Denoyelle S, He X, Zvereva N, Supko JG, Chorev M, *et al.* (2011) Chemical genetics identify eIF2α kinase heme-regulated inhibitor as an anticancer target. *Nat Chem Biol* 7: 610–616

Conway AJ, Brown FC, Fullinfaw RO, Kile BT, Jane SM and Curtis DJ (2017) A mouse model of hereditary coproporphyria identified in an ENU mutagenesis screen. *DMM Dis Model Mech* 10: 1005–1013

Gray LT, Puig Lombardi E, Verga D, Nicolas A, Teulade-Fichou MP, Londoño-Vallejo A and Maizels N (2019) G-quadruplexes Sequester Free Heme in Living Cells. *Cell Chem Biol* 26: 1681-1691.e5

Grow EJ, Flynn RA, Chavez SL, Bayless NL, Wossidlo M, Wesche DJ, Martin L, Ware CB, Blish CA, Chang HY, *et al.* (2015) Intrinsic retroviral reactivation in human preimplantation embryos and pluripotent cells. *Nature* 522: 221

Handschin C, Lin J, Rhee J, Peyer AK, Chin S, Wu PH, Meyer UA and Spiegelman BM (2005) Nutritional regulation of hepatic heme biosynthesis and porphyria through PGC-1alpha. *Cell* 122: 505–515

Kim HJ, Khalimonchuk O, Smith PM and Winge DR (2012) Structure, function, and assembly of heme centers in mitochondrial respiratory complexes. *Biochim Biophys Acta* 1823: 1604–1616

Lämsä V, Levonen AL, Sormunen R, Yamamoto M and Hakkola J (2012) Heme and heme biosynthesis intermediates induce heme oxygenase-1 and cytochrome P450 2A5, enzymes with putative sequential roles in heme and bilirubin metabolism: different requirement for transcription factor nuclear factor erythroid- derived 2-like 2. *Toxicol Sci* 130: 132–144

Lindberg RLP, Porcher C, Grandchamp B, Ledermann B, Bürki K, Brandner S, Aguzzi A and Meyer UA (1996) Porphobilinogen deaminase deficiency in mice causes a neuropathy resembling that of human hepatic porphyria. *Nat Genet* 12: 195–199

Magness ST, Maeda N and Brenner DA (2002) An exon 10 deletion in the mouse ferrochelatase gene has a dominant-negative effect and causes mild protoporphyria. *Blood* 100: 1470–1477

Nakamura T, Okamoto I, Sasaki K, Yabuta Y, Iwatani C, Tsuchiya H, Seita Y, Nakamura S, Yamamoto T and Saitou M (2016) A developmental coordinate of pluripotency among mice, monkeys and humans. *Nature* 537: 57–62

Pakos‐Zebrucka K, Koryga I, Mnich K, Ljujic M, Samali A and Gorman AM (2016) The integrated stress response. *EMBO Rep* 17: 1374–1395

Phillips JD, Jackson LK, Bunting M, Franklin MR, Thomas KR, Levy JE, Andrews NC and Kushner JP (2001) A mouse model of familial porphyria cutanea tarda. *Proc Natl Acad Sci* 98: 259–264

Renčiuk D, Ryneš J, Kejnovská I, Foldynová-Trantírková S, Andäng M, Trantírek L and Vorlíčková M (2017) G-quadruplex formation in the Oct4 promoter positively regulates Oct4 expression. *Biochim Biophys acta Gene Regul Mech* 1860: 175–183

Schmidt EK, Clavarino G, Ceppi M and Pierre P (2009) SUnSET, a nonradioactive method to monitor protein synthesis. *Nat Methods* 6: 275–277

Sperber H, Mathieu J, Wang Y, Ferreccio A, Hesson J, Xu Z, Fischer KA, Devi A, Detraux D, Gu H, *et al.* (2015) The metabolome regulates the epigenetic landscape during naive-to-primed human embryonic stem cell transition. *Nat Cell Biol* 17: 1523–1535

Di Stefano B, Ueda M, Sabri S, Brumbaugh J, Huebner AJ, Sahakyan A, Clement K, Clowers KJ, Erickson AR, Shioda K, *et al.* (2018) Reduced MEK inhibition preserves genomic stability in naive human embryonic stem cells. *Nat Methods* 15: 732–740

Taubenschmid-Stowers J, Rostovskaya M, Santos F, Ljung S, Argelaguet R, Krueger F, Nichols J and Reik W (2022) 8C-like cells capture the human zygotic genome activation program in vitro. *Cell Stem Cell* 29: 449-459.e6

Vinchi F, Ingoglia G, Chiabrando D, Mercurio S, Turco E, Silengo L, Altruda F and Tolosano E (2014) Heme exporter FLVCR1a regulates heme synthesis and degradation and controls activity of cytochromes P450. *Gastroenterology* 146: 1325–1338

Ware CB, Nelson AM, Mecham B, Hesson J, Zhou W, Jonlin EC, Jimenez-Caliani AJ, Deng X, Cavanaugh C, Cook S, *et al.* (2014) Derivation of naive human embryonic stem cells. *Proc Natl Acad Sci* 111: 4484–4489

Zhang X, Guo J, Wei X, Niu C, Jia M, Li Q and Meng D (2018) Bach1: Function, Regulation, and Involvement in Disease. *Oxid Med Cell Longev* 2018

Reviewer #1 (Recommendations for the authors):The authors will find below a list of my concerns.1. For readers of the stem cell field a more detailed introduction into the heme synthesis pathway needs to be included in the introduction.

A description of the heme synthesis pathway is now included in the introduction section of the revised version, line 65-67. The following has been added: “This pathway, starting in mitochondria, uses succinyl-CoA and glycine as starting material. It then proceeds to successive cytosolic reactions before ending by the formation of the heme molecule in the mitochondrial matrix.”

2. The statement made in lines 150-152 and shown as Suppl. Figure 2b should be further supported by proper quantification using replicate assays.

While further quantification of gene expression (*Cdx2, Tead4, Gata3, Eomes, Hand1, Elf5, Krt18, Cited1, Tfap2c and Ets1*) following SA or AA5 treatment to validate the increase in lineage potential still shows an increase (Author response image 3), we observed high variability and the increase in the expression of trophoblast-specific genes was not significant (n=6). While this still could be interesting, we hypothesize that since the number of 2CLCs is increased from 1 to 3 % upon SA or AA5 treatment, the effect on lineage potential will remain limited (see also the response to the Editor’s question 7). Because of the lack of significance, we removed the data and related discussion on the extended potential from the revised manuscript.

3. Lines 207-209. The rationale and the conclusion behind the experiment shown in Suppl. Figure 4 must be better explained.

The lines 251-254 of the manuscript have been amended to clarify the conclusion of Suppl. Figure 4 (now Figure 5 – Supplement Figure 1 in the revised manuscript). The following is now stated: “Of these three classes, we ruled out a role for PHDs, responsible for HIF1α degradation, as the abundance of this transcription factor was not increased by SA nor by AA5 treatment, demonstrating a lack of stabilization following a putative PHD inhibition (Figure 5 – Supplement Figure 1).”

4. Line 98: the word 'prevented' is too strong for the observed effect. Oct6, Otx2 and Dnmt3b are clearly upregulated, merely less than in WT.

We updated the line 98 to more accurately phrase the observed effect.

5. The authors could put their findings in context of data on the impact of modulating the intracellular αKG/succinate ratio on pluripotency maintenance (https://doi.org/10.1038/nature13981; https://doi.org/10.15252/embj.201899518).

We amended the end of the Discussion section to include the missing reference and commented further on the difference between these studies and ours.

6. Does BM also rescue the exit from pluripotency phenotype?

We thank reviewer 1 for this suggestion. Indeed, addition of BM to SA during the naïve state exit rescues the observed phenotype, revealing succinate as a mechanistic actor. This is now shown in the updated figure 2 (Figure 2f).

7. Quantification and comparison of pan succinate levels in Figures4c and 5a would be helpful and required for interpretation. As there is basically no staining detectable in untreated cells in 4a and some in 5a, it is unclear whether SA and AA5 effects are similar in size. It appears that AA5 treatment has a much weaker effect.

In order to further back our claims, we quantified the pan-succinyllysine residue abundance through flow cytometry. These quantifications were added to figures 2g, 4c and 5a. The quantification strengthens the results observed by immunofluorescence.

Reviewer #2 (Recommendations for the authors):– In the first part, regarding the analysis of previously published results of CRISPR-Cas9 screens (lines 77-87): It would be ideal to use an alternative annotation tool for gene ontology analyses. DAVID is already outdated. If the finding is corroborated with an alternative method, it not only gives strength to the initial observations but allows finding new targets of interest.

We compared the results generated by DAVID with other functional annotation tools. We ran over-representation analysis using the R package ClusterProfiler (Yu et al., 2012), which assess enrichment of terms by calculating *p*-values using the hypergeometric distribution method described in Boyle et al., 2004. The analysis was performed using GO databases (including biological process, molecular function and subcellular localization) as well as hallmark (h) (Liberzon et al., 2015) and canonical pathways (CP, part of curated database, C) databases of Molecular Signature Database (MSigDB, please see https://www.gsea-msigdb.org/). The results show significant enrichment of terms related to heme metabolism (“heme biosynthetic process”, “porphyrin-containing compound biosynthetic process”, “protoporphyrinogen IX metabolic process”) for both the human and mouse CRISPR-Cas9 screen list (Author response Tables 1 and 2) confirming the results generated by DAVID. We maintain the DAVID results in the main figure as the DAVID tool has been continuously updated and improved from initially reported publications (Jiao et al., 2012; Sherman et al., 2007, 2022). As of today, the last update of the DAVID knowledgebase (v2023q1) was released in April 2023 and is cited in more than 60 000 scientific publications highlighting its robustness.

**Author response table 1. sa2table1:** Overrepresentation analysis (ORA) of mESC screen. Top 50 hit of the biological process gene ontology (GO) terms from the mouse CRISPR-Cas9 screen GO:0140053 mitochondrial RNA metabolic process

ID	Description	GeneRatio	BgRatio	pvalue	p.adjust
mitochondrial gene expression	59/530	108/23355	2,1943E-68	8,1912E-65
GO:0032543	mitochondrial translation	50/530	76/23355	5,4613E-64	1,0194E-60
GO:0033108	mitochondrial respiratory chain complex assembly	47/530	84/23355	2,9765E-55	3,7037E-52
GO:0010257	NADH dehydrogenase complex assembly	37/530	49/23355	3,0091E-51	2,2466E-48
GO:0032981	mitochondrial respiratory chain complex I assembly	37/530	49/23355	3,0091E-51	2,2466E-48
GO:0007005	mitochondrion organization	79/530	489/23355	4,5039E-44	2,8022E-41
GO:0006091	generation of precursor metabolites and energy	59/530	404/23355	1,7562E-30	9,3653E-28
GO:0045333	cellular respiration	40/530	159/23355	2,1561E-30	1,0061E-27
GO:0022900	electron transport chain	30/530	84/23355	3,6581E-28	1,5173E-25
GO:0022904	respiratory electron transport chain	29/530	80/23355	1,7473E-27	6,5225E-25
GO:0046034	ATP metabolic process	43/530	256/23355	6,2247E-25	2,1125E-22
GO:0006119	oxidative phosphorylation	30/530	107/23355	1,3283E-24	4,1322E-22
GO:0015980	energy derivation by oxidation of organic compounds	42/530	250/23355	2,222E-24	6,3806E-22
GO:0042773	ATP synthesis coupled electron transport	24/530	62/23355	9,046E-24	2,4121E-21
GO:0042775	mitochondrial ATP synthesis coupled electron transport	23/530	58/23355	4,023E-23	1,0012E-20
GO:0000959	18/530	45/23355	1,8641E-18	4,3491E-16
GO:0043039	tRNA aminoacylation	17/530	43/23355	2,1272E-17	4,671E-15
GO:0043038	amino acid activation	17/530	44/23355	3,3946E-17	7,0399E-15
GO:0006399	tRNA metabolic process	26/530	161/23355	3,9816E-15	7,8229E-13
GO:0006418	tRNA aminoacylation for protein translation	15/530	40/23355	4,2779E-15	7,9846E-13
GO:0017004	cytochrome complex assembly	14/530	34/23355	7,4415E-15	1,3228E-12
GO:0006120	mitochondrial electron transport, NADH to ubiquinone	12/530	22/23355	8,676E-15	1,4722E-12
GO:0008535	respiratory chain complex IV assembly	12/530	24/23355	3,4812E-14	5,6501E-12
GO:0070129	regulation of mitochondrial translation	12/530	27/23355	2,1033E-13	3,2715E-11
GO:0006744	ubiquinone biosynthetic process	10/530	17/23355	5,6158E-13	8,0629E-11
GO:1901663	quinone biosynthetic process	10/530	17/23355	5,6158E-13	8,0629E-11
GO:0006520	cellular amino acid metabolic process	28/530	244/23355	2,2261E-12	3,0777E-10
GO:0062125	regulation of mitochondrial gene expression	12/530	32/23355	2,4637E-12	3,2847E-10
GO:0006743	ubiquinone metabolic process	10/530	19/23355	2,5607E-12	3,2962E-10
GO:0034660	ncRNA metabolic process	38/530	445/23355	2,942E-12	3,6609E-10
GO:0006401	RNA catabolic process	28/530	258/23355	8,6577E-12	1,0426E-09
GO:0033617	mitochondrial cytochrome c oxidase assembly	10/530	21/23355	9,3856E-12	1,0949E-09
GO:0009060	aerobic respiration	16/530	76/23355	1,2209E-11	1,3811E-09
GO:0034248	regulation of cellular amide metabolic process	35/530	409/23355	2,0566E-11	2,258E-09
GO:0006417	regulation of translation	32/530	350/23355	2,7541E-11	2,9374E-09
GO:0006783	heme biosynthetic process	10/530	23/23355	2,9224E-11	2,9937E-09
GO:0032259	methylation	32/530	351/23355	2,9672E-11	2,9937E-09
GO:0007034	vacuolar transport	20/530	143/23355	9,0443E-11	8,8848E-09
GO:0006402	mRNA catabolic process	24/530	220/23355	2,4302E-10	2,3261E-08
GO:0006779	porphyrin-containing compound biosynthetic process tetrapyrrole biosynthetic process	10/530	28/23355	3,0269E-10	2,7019E-08
GO:0033014		10/530	28/23355	3,0269E-10	2,7019E-08
GO:0043414	macromolecule methylation	28/530	300/23355	3,0399E-10	2,7019E-08
GO:0019827	stem cell population maintenance	21/530	172/23355	4,0678E-10	3,5106E-08
GO:0000956	nuclear-transcribed mRNA catabolic process	17/530	109/23355	4,1379E-10	3,5106E-08
GO:0098727	maintenance of cell number	21/530	176/23355	6,2463E-10	5,1816E-08
GO:0016569	covalent chromatin modification	35/530	464/23355	6,4242E-10	5,2133E-08
GO:0016570	histone modification	34/530	450/23355	1,0876E-09	8,6383E-08
GO:0042168	heme metabolic process	10/530	32/23355	1,3714E-09	1,0666E-07
GO:0032008	positive regulation of TOR signaling	11/530	43/23355	2,2114E-09	1,6847E-07

**Author response table 2. sa2table2:** Overrepresentation analysis (ORA) of hESC screen. Top 50 hit of the biological process gene ontology (GO) terms from the human CRISPR-screen

ID	Description	GeneRatio	BgRatio	pvalue	p.adjust
GO:0097194	execution phase of apoptosis	8/160	94/18862	1,3494E-06	0,00187098
GO:0006414	translational elongation	9/160	134/18862	2,1432E-06	0,00187098
GO:0032543	mitochondrial translation	9/160	134/18862	2,1432E-06	0,00187098
GO:0090200	positive regulation of release of cytochrome c from mitochondria	5/160	27/18862	2,8639E-06	0,00187515
GO:0070125	mitochondrial translational elongation	7/160	88/18862	9,8885E-06	0,00453049
GO:0140053	mitochondrial gene expression	9/160	165/18862	1,1757E-05	0,00453049
GO:2001235	positive regulation of apoptotic signaling pathway	8/160	126/18862	1,2109E-05	0,00453049
GO:0097193	intrinsic apoptotic signaling pathway	11/160	283/18862	3,0718E-05	0,0099803
GO:0090199	regulation of release of cytochrome c from mitochondria	5/160	44/18862	3,4297E-05	0,0099803
GO:0051204	protein insertion into mitochondrial membrane	5/160	48/18862	5,2618E-05	0,01378068
GO:1900739	regulation of protein insertion into mitochondrial membrane involved in apoptotic signaling pathw	4/160	26/18862	6,445E-05	0,01406611
GO:1900740	positive regulation of protein insertion into mitochondrial membrane involved in apoptotic signali	4/160	26/18862	6,445E-05	0,01406611
GO:0090151	establishment of protein localization to mitochondrial membrane	5/160	53/18862	8,5226E-05	0,01586063
GO:0046501	protoporphyrinogen IX metabolic process	3/160	11/18862	9,4023E-05	0,01586063
GO:0006919	activation of cysteine-type endopeptidase activity involved in apoptotic process	6/160	87/18862	9,7175E-05	0,01586063
GO:0001836	release of cytochrome c from mitochondria	5/160	55/18862	0,00010191	0,01586063
GO:0006400	tRNA modification	6/160	89/18862	0,00011028	0,01586063
GO:0070126	mitochondrial translational termination	6/160	89/18862	0,00011028	0,01586063
GO:0001844	protein insertion into mitochondrial membrane involved in apoptotic signaling pathway	4/160	30/18862	0,00011506	0,01586063

– The second title, "Heme synthesis inhibition prevents the activation of key developmental signaling pathways," needs to be more specific. Because heme is a critical component of general cell homeostasis, it is evident that the inhibition will affect key signaling pathways, not only in development but in any stage of mammal's life.In this same section, what happens with positive enriched genes in the SA-treated condition? It should be added the up-regulation plot side Figure 2a, and a correspondent discussion about it.

We thank reviewer 2 for these comments. We accordingly corrected the section 2 as a whole, changing its title and the related figure 2a. The new secondary title is Heme synthesis inhibition prevents the activation of key signaling pathways associated with implantation. The upregulated terms have been also commented in the main text section (lines 132-134).

– In the section describing the upregulation of 2-cell-stage-related genes when SA is added, it is needed to add a scramble-inhibition as control. Treat 2iL cells with an inhibitor of a different metabolic pathway for 48h and analyze by rt-PCR (a) the 2-cell-related genes, (b) the differentiation towards trophoblast cells.

In order to validate the metabolic effect of heme synthesis inhibition we inhibited two other unrelated metabolic pathways: the pentose phosphate pathway with 6-aminonicotinamide and the fatty acid oxidation with Etoxomir. We do not observe any effect of the inhibition of these pathways on the upregulation of the 2CLC program (Figure 1 - Supplement Figure 1c).

– What happens with the glycine levels after heme synthesis inhibition? Does glycine play a role in the induction of a 2-cell-like state? Ideally authors should include an experiment to test this role.

To investigate the putative role of a glycine accumulation on the acquisition of a 2CLC gene signature, we tested the supplementation in 10 mM of glycine of the 2iL naïve medium. As shown in Author response image 5, no gene expression increase is observed.

**Author response image 5. sa2fig5:** Glycine supplementation does not trigger a 2CLC gene signature. Relative expression of 2C gene markers of mESCs assessed by RT-qPCR relative to Gapdh expression and to 2iL naïve control, after supplementation with 10 mM of glycine for 48 h. n=3 independent replicates. Data shown as means +/-S.D. p>0.05, T-tests.

– The following is optional, but it could be a considerable improvement to provide the bulk RNA seq comparing the inhibition of SUCNR1 vs. the activation induced by succinate, to see all other pathways that could crosstalk and describe better the mechanism of the succinate in pluripotent/totipotent states.

While comparing the effect we observe on the 2CLC gene signature through heme synthesis or SDH inhibition to the previously published RA induction we noticed that inhibition of RAR and RXR rescued the increase in gene expression (Figure 5c). While this does not pinpoint the exact mechanism, these observations bring light to an intermediate involved.

– The authors provide a very detailed molecular analysis on the effect of heme synthesis inhibition on the pluripotency exit. However, functional characterization (such as teratoma formation) is important to tell whether these naive mESC after treatment with heme synthesis blocker have any defects on the pluripotency exit and differentiation.

As explained in the Discussion section, the fact that heme synthesis knock-out is embryonic lethal around the implantation stage in mouse (Lindberg *et al.,* 1996; Bensidhoum *et al.,* 1998; Phillips *et al.,* 2001; Conway *et al.,* 2017; Magness *et al.,* 2002) strongly suggests an implantation defect, hence, pluripotency exit. This is further supported by the loss of Oct4 in ALAD KO mESCs after removal of hemin (Figure 1 – Supplement Figure 1b). Furthermore, teratoma formation assay or embryoid body formation assays after a temporary blockade (2 days) of heme synthesis through SA would quickly be attenuated as heme synthesis would resume after injection, especially since these assays extends for weeks.

– Although it is clear that heme synthesis is important for the mouse pluripotency status transition, it is not clear whether this is a mechanism present in human as well. The authors presented bioinformatic data suggesting that this might be the case. It is important that the authors perform functional studies in the human cells to further substantiate this point.

To answer this question, we used Elf1 hESC cells (Ware *et al.,* 2014) grown in RSeT media to mimic the naïve conditions, and triggered the exit toward the primed stage for 4 days in TeSR media (both commercially available, StemCell technologies), with or without 0.5 mM of SA. We observe a significant prevention of the loss of naïve markers, as in mESCs, while not preventing the increase in a primed signature (Figure 1 – Supplement Figure 1c).

– It is not clear how the cell senses heme deficiency and how heme deficiency results in inability to activate ERK/MAPK and TGF β signaling. Could the authors provide comments into this aspect?

We thank the reviewers for their highly relevant comments. We have addressed these two questions simultaneously.

Heme deficiency is canonically sensed in mammals through the nuclear factor BACH1 (reviewed in (Zhang *et al.,* 2018)) or the activation of the Integrated Stress Response (ISR) through the Heme Responsive Inhibitor (HRI) or EIF2AK1 (reviewed in (Pakos‐Zebrucka *et al.,* 2016)).

BACH1 heterodimerized with MAF proteins (among others) binds to DNA and represses the transcription of target genes. Upon heme binding to BACH1, the complex dissociates and BACH1 is exported in the cytosol, thus relieving the repression of target genes. As seen in immunofluorescence analyses of BACH1 subcellular localization (Figure 2 – Supplement Figure 1a), the abundance of nuclear BACH1 is comparable in 2iL, EpiLC, and EpiLC+SA conditions, indicating that the inhibition of heme synthesis does not affect BACH1 activity. On the opposite and as expected, the addition of hemin provokes the nuclear exclusion of BACH1. This data indicates that BACH1 is not responsible for the heme deficiency sensing in our model.

As another putative way of heme depletion to act on the transition from naïve to primed, we also investigated the role of the ISR. Indeed, various environmental and pathological conditions, including protein homeostasis (proteostasis) defects, nutrient deprivation, viral infection, and oxidative stress, activate the ISR in order to restores balance by reprogramming gene expression, all converging to the phosphorylation and activation of the eukaryotic translation initiation factor eIF2. Heme depletion is known to be one of the stresses activating this pathway, allowing the phosphorylation and activation of the heme-regulated inhibitor (HRI) kinase, in turn phosphorylating EIF2α. To assess the activation of this pathway, we targeted two levels: The phosphorylation of EIF2α by western blot analysis and the global protein synthesis levels with a SUnSET assay (Schmidt *et al.,* 2009). The increase in EIF2α phosphorylation (Figure 2 – Supplement Figure 1b), a direct result of the activity of HRI in absence of heme, indicates the activation of a kinase upstream of the pathway. The global reduction in protein synthesis, observed with the levels of puromycin-labelled peptides (Figure 2 – Supplement Figure 1c), further confirm the activation of the ISR upon inhibition of heme synthesis.

To further explore the possible implication of this ISR-HRI axis in the inhibition of native-to-primed transition induced by heme synthesis inhibition, we used a chemical activator of HRI (BTdCPU; (Chen *et al.,* 2011)) to assess whether it could induce a similar effect as SA or not. Treatment with 2 µM BTdCPU, reduces protein synthesis to a similar extent that SA (Figure 2 – Supplement Figure 1d), but fails at preventing the naïve stage exit (Figure 2 – Supplement Figure 1e), indicating an HRI-independent mechanism. These results were added to the manuscript in lines 119-126.

– How strong the effect of inhibiting heme synthesis and succinate in inducing the 2CLC population are as the absolute number of Zscan4+ or MuERVL+ cell are quite low (< 3%) ?. I suggest that the authors include other 2CLC inducers such as acetate(Rodriguez-Terrones, 2020) and RA (Iturbide, A., 2021) as a control for comparison.

The updated figure 5 now includes RA as a positive control as presented in Iturbide et al., 2021 and as suggested by the reviewers. Interestingly, we observe that the gene expression levels of the 2CLC markers after RA treatment increase to a similar extent than with AA5, but not SA.

These authors showed that the RA-induced 2CLC increase is due to activation of the RARγ since inhibition of RXR (with the HX531 inhibitor) does not rescue the effect while a RAR inhibitor (AGN193109) does. Very interestingly, the AA5-induced 2CLC signature seems to depend on both receptors as HX531 (RXR inhibitor) and AGN193109 (pan-RAR inhibitor) do inhibit the AA5-induced 2CLC gene expression signature.

Since genetic ablation of heme synthesis enzymes is embryonic lethal around the implantation time (Lindberg *et al.,* 1996; Bensidhoum *et al.,* 1998; Phillips *et al.,* 2001; Conway *et al.,* 2017; Magness *et al.,* 2002), we expect this experiment to bear toxicity for the embryo, especially since accumulation of heme synthesis intermediate is known for its toxicity (Lämsä *et al.,* 2012; Handschin *et al.,* 2005). Furthermore, injection of the 2CLC in early embryos is rarely used, including in Iturbide (2021) or Rodriguez-Terrones (2020).

– The authors performed the 2CLC experiment with 2iL condition. However, a quick review of the literature indicates that many of these studies have been done in FBS/LIF system. Can the authors comment on the difference between 2iL condition and FBS/LIF in the study of 2CLC?

We thank the reviewer to point at this difference. We also observe an increase in the expression of 2CLC marker genes in response to AA5, followed by a reduction when the FBS+LIF basal media was supplemented with BM (Author response image 6). This finding highlights a conserved mechanism with the 2iLIF basal media.

**Author response image 6. sa2fig6:** Succinate accumulation induces a 2CLC gene signature in mESCs grown in FBS+LIF conditions. Gene expression analysis of 2CLC genes of mESCs grown in a media with 15% FBS supplemented with LIF, with or without addition of 250nM atpenin A5 and 10µM butylmalonate, assessed by RT-qPCR relative to Gapdh expression and to FBS/LIF control. Results expressed as mean +/- S.D. *p<0.05, **p < 0.01, ***p < 0.001. ANOVA-1. n=3 independent biological replicates.

– The authors analyzed that heme synthesis is also important human pluripotent state transition. It will be interesting to see the scenario in human pluripotent stem cells, which will largely make the study difference, especially if the heme synthesis blockade will also increase the emergence of human 2 cell-like cells.

To answer this question, we used Elf1 hESC cells (Ware *et al.,* 2014) grown in RSeT media to mimic the naïve conditions, and triggered the exit toward the primed stage for 4 days in TeSR media (both commercially available, StemCell technologies), with or without 0.5 mM of SA. We observe a significant prevention of the loss of naïve markers, as in mESCs, while not preventing the increase of a primed signature (Figure 1 – Supplement Figure 1c).

In addition, naïve Elf1 hESCs maintained either in RSeT medium or in 2iL-I-F (GSK3 and MEK inhibitors, LIF, IGF1 and FGF2) media (Sperber *et al.,* 2015) treated with 0.5 mM of SA for 2 days, show a significant upregulation of markers associated with the 8-cell stage (the human homolog of the 2CLC), corresponding to the zygote genome activation event (Taubenschmid-Stowers *et al.,* 2022), highlighting a similar mechanism (Author response image 4; Figure 3g).

– The whole study of inhibition is chemical achieved. As a supplement, the authors should also test with genetic intervention instead of sole chemicals. For example, the inhibition of heme synthesis by SA should also be confirmed by knockout or knockdown of proteins in this pathway like ALAD, PBGD et al. Also similar with AA5.

We thank the reviewer for this important comment. Using the CRISPR-Cas9 technology we generated an ALAD KO E14-mESC line using CRISPR/Cas9 (Figure 1 – Supplement Figure 1a). To maintain a healthy population, this line was grown in a medium supplemented with hemin and removal of this supplement allowed to reveal the heme synthesis inhibition. As shown in the updated figure 3 (Figure 3e), removal of hemin in the culture medium for 2 days induces a strong increase in 2CLC marker expression.

We also performed the transition from naïve to primed in these ALAD KO mESCs. Genetic ablation of ALAD also prevents mESCs to properly exit the naïve state as the naïve marker expression is maintained. On the other hand, primed markers, except Dnmt3b, are upregulated (Figure 1 – Supplement Figure 1b). While this is in apparent contrast to the chemical inhibition of ALAD, the complete loss of ALAD seems to perturb the pluripotency state as seen by the complete loss of Oct4 expression, thus impairing the proper transition. This loss of Oct4 transcript could be due to the complete loss of heme, impairing the formation of G-quadruplexes, known to promote Oct4 gene expression (Renčiuk *et al.,* 2017; Gray *et al.,* 2019).

– The functional assay of 2 cell-like cells after heme synthesis inhibition should be tested in vivo by 8-cell or blastocyst injection– To make the finding here more convincing and significant, it will be good to confirm in embryos.

As explained in the Discussion section, the fact that heme synthesis knock-out is embryonic lethal around the implantation stage in mouse (Lindberg *et al.,* 1996; Bensidhoum *et al.,* 1998; Phillips *et al.,* 2001; Conway *et al.,* 2017; Magness *et al.,* 2002) strongly suggests an implantation defect, hence, pluripotency exit. This is further supported by the loss of Oct4 in ALAD KO mESCs after removal of hemin (Supp. Figure 1A). While we expect either reduction in survival or development arrest by the treated embryos (heme synthesis or SDH inhibition), obtaining embryos for the experiments was challenging and thus was not performed.

– The manuscript should include the metabolism part in the introduction instead of using large paragraphs in the main text to introduce these pathways and previous studies.

We updated the introduction section in lines 65-67. However, the bulk of the heme synthesis/succinate flow remained in the result section to better guide the reader.